# Beyond Heuristics: Learnable Density Control for 3D Gaussian Splatting

**Zhenhua Ning** [1 2]  **Xin Li** [1 †]  **Jun Yu** [2]  **Guangming Lu** [2]  **Yaowei Wang** [2 1]  **Wenjie Pei** [2 1 †]

## Abstract

While 3D Gaussian Splatting (3DGS) has demonstrated impressive real-time rendering performance, its efficacy remains constrained by a reliance on heuristic density control. Despite numerous refinements to these handcrafted rules, such methods inherently lack the flexibility to adapt to diverse scenes with complex geometries. In this paper, we propose a paradigm shift for density control from rigid heuristics to fully learnable policies. Specifically, we introduce **LeGS**, a framework that reformulates density control as a parameterized policy network optimized via Reinforcement Learning (RL). Central to our approach is the tailored effective reward function grounded in sensitivity analysis, which precisely quantifies the marginal contribution of individual Gaussians to reconstruction quality. To maintain computational tractability, we derive a closed-form solution that reduces the complexity of reward calculation from $O(N^2)$ to $O(N)$. Extensive experiments on the Mip-NeRF 360, Tanks & Temples, and Deep Blending datasets demonstrate that **LeGS** significantly outperforms state-of-the-art methods, striking a superior balance between reconstruction quality and efficiency. The code will be released at https://github.com/AaronNZH/LeGS.

## 1. Introduction

3D Gaussian Splatting (3DGS) (Kerbl et al., 2023) has emerged as a premier technique for 3D scene reconstruction, leveraging 3D Gaussians to achieve high-fidelity scene representation alongside real-time rendering speeds. A pivotal step of 3DGS is adaptive density control, a mechanism that dynamically refines the Gaussian distribution—specifically their quantity, spatial density, and geometric parameters—to optimize the scene representation.

Most existing methods adopt a heuristic paradigm for adaptive density control centered on two fundamental operations: densification and pruning. Original 3DGS performs densification by cloning or splitting Gaussians that exhibit high positional gradients, while pruning ineffective Gaussians via thresholding of opacity values to maintain computational efficiency. Despite its simplicity, such rigid heuristics often fail to generalize across diverse and complex scenarios, frequently resulting in rendering artifacts such as blurriness and Gaussian redundancies. In light of this, recent works have investigated several refinements to this heuristic density control mechanism. Specifically, Pixel-GS (Zhang et al., 2024) normalizes positional gradients based on pixel coverage to alleviate under-reconstructions, while FastGS (Ren et al., 2025) targets regions exhibiting high rendering errors. Taming-3DGS (Mallick et al., 2024) and Perceptual-GS (Zhou & Ni, 2025) incorporate saliency and perceptual priors to emphasize visually significant Gaussians. While these methods have demonstrated substantial improvements in both representation quality and efficiency, a significant limitation of the heuristic density control paradigm remains: the reliance on handcrafted rules and predefined hyperparameters. Such rigid designs inherently constrain generalizability, as the optimal thresholds for densification and pruning often vary significantly across diverse scenes with complex geometric conditions and varying scales. As evidenced by Figure 1, the density control in FastGS is agnostic to regional geometry and texture, failing to adaptively allocate Gaussians according to the underlying geometric or textural complexity.

To circumvent the aforementioned limitations of heuristic density control, we propose an entirely different paradigm: Learnable Density Control for Gaussian Splatting (**LeGS**). In contrast to existing heuristic-driven frameworks, **LeGS** formulates density control as a learned policy parameterized by a neural network. By determining optimal refinement operations at each iteration in a data-driven manner, our approach enables superior adaptation to diverse scenes with complex geometries and textures. Specifically, by characterizing the iterative adaptive density control operations as a sequence of discrete actions—specifically *maintain*, *clone*, *split*, and *prune*—and treating Gaussian attributes as inter-

---

[1]Pengcheng Laboratory, Shenzhen [2]Harbin Institute of Technology, Shenzhen. Correspondence to: Wenjie Pei <wenjiecoder@outlook.com>, Xin Li <xinlihitsz@gmail.com>.

*Proceedings of the 43$^{rd}$ International Conference on Machine Learning*, Seoul, South Korea. PMLR 306, 2026. Copyright 2026 by the author(s).

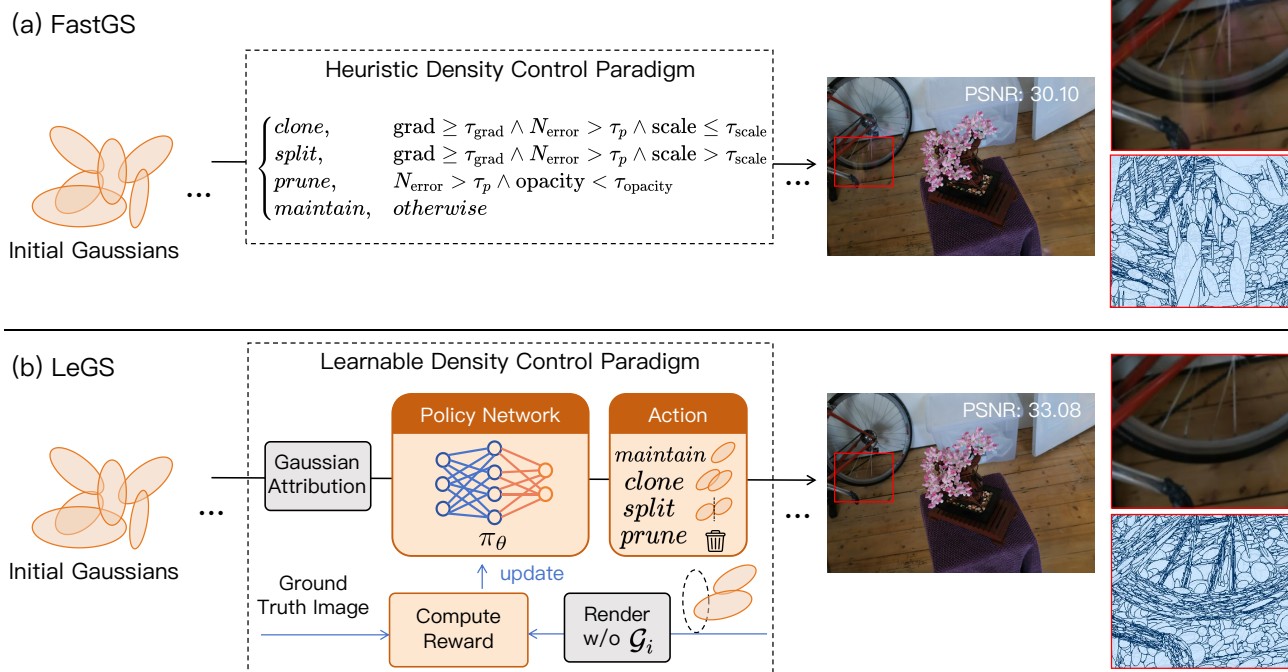

*Figure 1.* Comparison between our LeGS and FastGS. The heuristic paradigm, relying on a manually-designed score function with a fixed threshold, achieves suboptimal results. For instance, FastGS tends to under-densify Gaussians in wheel (red box), causing artifacts. In contrast, our learning-based paradigm effectively captures intricate local details.

active states, the optimization of **LeGS** can be naturally formulated as a Reinforcement Learning (RL) problem.

The design of the reward function is essential to the efficacy of the Reinforcement Learning process in terms of both the reconstruction fidelity and convergence efficiency. We draw inspiration from sensitivity analysis (Zeiler & Fergus, 2013) and design an effective reward function for our **LeGS**. The core idea is to evaluate the reconstruction impact of each Gaussian by measuring the sensitivity of the rendering quality to each Gaussian. Specifically, we quantify each Gaussian's contribution by aggregating its impact on rendering quality across its entire projected footprint. Nevertheless, a naive implementation of reward calculation is computationally prohibitive. Since evaluating the marginal contribution of each Gaussian necessitates a full rendering pass, the resulting $O(N^2)$ complexity—where $N$ is the number of Gaussians—imposes an unsustainable overhead on the training process. To address this, we derive a closed-form solution for the rendering output that simulates the ablation of individual Gaussians, leveraging intermediate rendering states. This efficient-yet-exact formulation circumvents the need for actual re-rendering, allowing us to compute the sensitivity in constant time ($O(1)$) per Gaussian. Consequently, the overall complexity is reduced to $O(N)$. Given that this reward is computed periodically (e.g., every 100 iterations), the cumulative computational overhead remains negligible. To conclude, the contributions of this work are summarized

as follows:

- We identify the inherent limitations of heuristic-based density control in 3DGS and propose **LeGS**, a novel learnable paradigm that reformulates density control as a parameterized policy network within a reinforcement learning (RL) framework. By replacing rigid handcrafted rules with a data-driven and optimization-driven strategy, LeGS dynamically optimizes Gaussian densification and pruning in a scene-adaptive and context-aware manner.

- We design a novel reward function grounded in sensitivity analysis to precisely quantify the marginal contribution of each Gaussian to reconstruction quality, thereby providing effective learning signals for reinforcement learning. Furthermore, we derive an exact and efficient formulation for reward computation, which avoids exhaustive re-rendering and reduces the computational complexity from quadratic $O(N^2)$ to linear $O(N)$.

- Extensive experiments demonstrate that our method achieves state-of-the-art performance on the Mip-NeRF 360, Tanks & Temples, and Deep Blending datasets. Qualitative results further show that our method yields reconstructions better aligned with scene textures. In addition, the efficiency analysis shows that our method offers a favorable trade-off between rendering efficiency and reconstruction quality.

## 2. Related Work

### 2.1. 3D Gaussian Splatting for Novel View Synthesis

Novel view synthesis aims to generate photorealistic images from arbitrary viewpoints given a set of calibrated input images. Neural radiance fields (NeRF) (Mildenhall et al., 2020) and subsequent variants (Barron et al., 2021; 2022; Verbin et al., 2024; Fridovich-Keil et al., 2022; Turki et al., 2022; Niemeyer et al., 2022; Poole et al., 2023; Ni et al., 2024; Xie et al., 2024) have achieved remarkable rendering quality by modeling scenes with implicit neural representations and differentiable volume rendering. However, their reliance on dense ray marching and volumetric integration often leads to substantial computational overhead, limiting their practicality for real-time or large-scale applications. To overcome these limitations, 3D Gaussian Splatting (3DGS) (Kerbl et al., 2023) introduces an explicit representation based on anisotropic 3D Gaussians, which can be rendered efficiently through differentiable rasterization. Thanks to its high rendering quality and real-time performance, 3DGS has rapidly inspired a wide range of follow-up studies. Recent works have extended 3DGS to dynamic scene reconstruction (Luiten et al., 2024; Wu et al., 2024; Yang et al., 2024; Duan et al., 2024), sparse-view and few-shot novel view synthesis (Zhu et al., 2024; Chen et al., 2024), surface reconstruction and geometry regularization (Huang et al., 2024; Dai et al., 2024), large-scale and urban scene modeling (Lin et al., 2024; Liu et al., 2024b; Lu et al., 2024), as well as compact and efficient representations (Lee et al., 2024; Liu et al., 2024a).

### 2.2. Density Control in 3DGS

3DGS employs a discrete Gaussian-based representation where reconstruction quality depends critically on density control—the adaptive allocation of Gaussians. The original 3DGS (Kerbl et al., 2023) manages density by monitoring view-space positional gradients, densifying regions through cloning or splitting based on Gaussian scale, while pruning ineffective Gaussians to maintain efficiency. Recent advances have refined these criteria along three directions. **Gradient-based refinements** address limitations in how positional gradients are computed. AbsGS (Ye et al., 2024) accumulates absolute gradient values to prevent gradient cancellation in blurry regions, while Pixel-GS (Zhang et al., 2024) weights gradients by pixel coverage to reduce blur artifacts. **Metric-driven strategies** introduce alternative scoring mechanisms for densification. RevisingGS (Rota Bulò et al., 2024) tracks maximum accumulated pixel error across views, FastGS (Ren et al., 2025) prioritizes Gaussians by average high-error pixel counts, TamingGS (Mallick et al., 2024) combines multiple scores through weighted aggregation, and Perceptual-GS (Zhou & Ni, 2025) leverages multi-view perceptual sensitivity to allocate Gaussians in visually salient regions. **Structural reformulations** rethink

the underlying optimization process: Mini-Splatting (Fang & Wang, 2024) employs depth-based re-initialization to mitigate Gaussian overlapping and under-reconstruction, while 3DGS-MCMC (Kheradmand et al., 2024) recasts density control as probabilistic sampling via Stochastic Gradient Langevin Dynamics. Despite these advances, existing methods fundamentally rely on heuristic rules with extensive hyperparameters, thereby achieving suboptimal performance.

## 3. Methodology

### 3.1. Preliminaries

3DGS represents the scene using an explicit set of anisotropic 3D Gaussians:

$$\left\{ \mathcal{G}_i(\mathbf{x}) = \exp\left(-\tfrac{1}{2}(\mathbf{x} - \boldsymbol{\mu}_i)^\top \boldsymbol{\Sigma}_{i_{3D}}^{-1}(\mathbf{x} - \boldsymbol{\mu}_i)\right) \right\}_{i=1}^{N}. \quad (1)$$

Each Gaussian $\mathcal{G}_i$ is parameterized by its mean position $\mu_i \in \mathbb{R}^3$, rotation $r_i \in \mathbb{R}^4$, scale $\mathcal{S}_i \in \mathbb{R}^3$, opacity $\sigma_i \in \mathbb{R}$, and color coefficients $\mathbf{c}_i \in \mathbb{R}^{16 \times 3}$ encoded via Spherical Harmonics (SH). The rotation $r_i$ and scale $\mathcal{S}_i$ are used to construct the 3D covariance matrix $\boldsymbol{\Sigma}_{i_{3D}}$. 3DGS synthetic image via $\alpha$-blending. For a set of Gaussians $\{\mathcal{G}_i\}_{i=1}^{N}$ sorted by depth, the color of pixel $p$ is accumulated as:

$$C(p) = \sum_{i=1}^{N} c_i \alpha_i \prod_{j=1}^{i-1} (1 - \alpha_j), \quad \alpha_i = \sigma_i \mathcal{G}_i'(p), \quad (2)$$

where

$$\mathcal{G}_i'(p) = \exp\left(-\tfrac{1}{2}(p - \boldsymbol{\mu}_{i_{2D}})^\top \boldsymbol{\Sigma}_{i_{2D}}^{-1}(p - \boldsymbol{\mu}_{i_{2D}})\right), \quad (3)$$

with $\boldsymbol{\mu}_{i_{2D}}$ and $\boldsymbol{\Sigma}_{i_{2D}}$ denoting the mean and covariance of the projected 2D Gaussian, respectively.

**Heuristic Paradigm for Adaptive Density Control.** In the original 3DGS implementation (Kerbl et al., 2023), Gaussians with accumulated view-space positional gradients exceeding a threshold $\tau_{grad}$ are identified as under-densification. Subsequently, these Gaussians are cloned if their scale is smaller than $\tau_{scale}$, and split otherwise. In parallel, 3DGS prunes Gaussians with opacity below $\tau_{opacity}$, as they contribute minimally to the final rendering. These heuristic strategies can be formulated as a function:

$$\mathcal{F} = \begin{cases} clone, & \text{grad} \geq \tau_{grad} \wedge \text{scale} \leq \tau_{scale} \\ split, & \text{grad} \geq \tau_{grad} \wedge \text{scale} > \tau_{scale} \\ prune, & \text{opacity} < \tau_{opacity} \\ maintain, & \text{otherwise} \end{cases} \quad (4)$$

### 3.2. Learnable Density Control Paradigm

To transition from heuristics to a learning-based paradigm, we parameterize the function $\mathcal{F}$ as a policy network $\mathcal{F}_\theta$ :

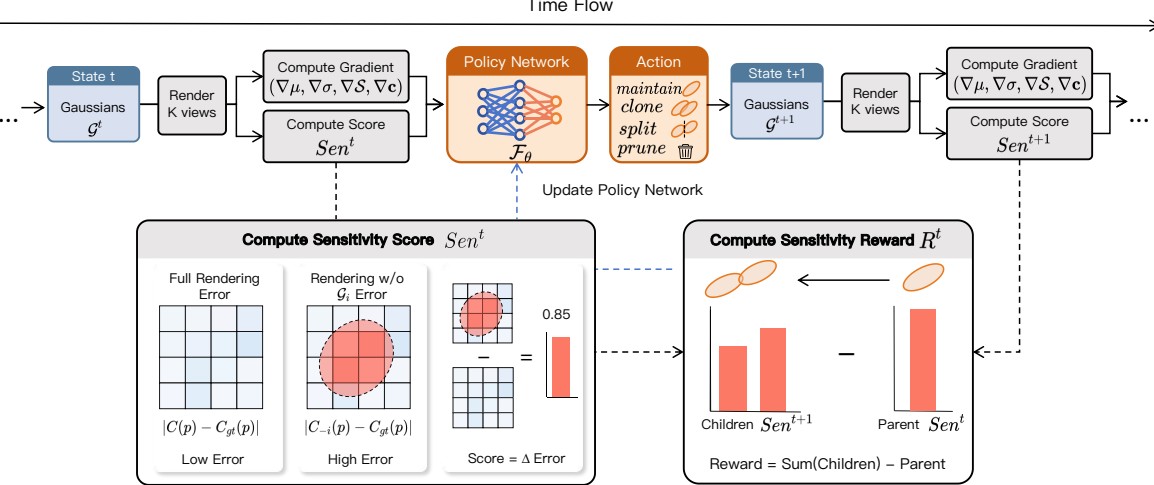

*Figure 2.* Overview of the LeGS framework. The pipeline renders $K$ views, computes Gaussian gradients ($\nabla\mu, \nabla\sigma, \nabla S, \nabla c$) and sensitivity scores as policy network inputs. The policy network $\mathcal{F}_\theta$ outputs actions (*maintain, clone, split, prune*) to update Gaussians. Sensitivity scores measure individual Gaussian rendering contributions, while sensitivity rewards quantify improvements from operations, enabling policy optimization via reinforcement learning.

$\mathbb{R}^d \to \mathbb{R}^4$, where $\theta$ denotes the learnable parameters. An overview of our proposed framework is depicted in Figure 2. Let $\mathcal{G}^t$ denote the Gaussians prior to the density control step at iteration $t$. We render $K$ views to compute the gradients and sensitivity scores $\text{Sen}^t$, which serve as inputs to the policy network $\mathcal{F}_\theta$. This network generates probabilities of actions $\mathcal{A}^t \in \mathbb{R}^4$, which govern the densification and pruning operations. These operations, in turn, yield the updated Gaussians $\mathcal{G}^{t+1}$. Subsequently, these updated Gaussians $\mathcal{G}^{t+1}$ are utilized in the following optimization process.

By characterizing the adaptive density control operations as a sequence of discrete actions and treating Gaussians as states, the optimization of **LeGS** is naturally formulated as an RL problem. Consequently, with an appropriate reward function, we can optimize the policy network using an RL algorithm. Specifically, we employ sensitivity analysis (Zeiler & Fergus, 2013) to calculate sensitivity scores $\text{Sen}^t$ and $\text{Sen}^{t+1}$ for Gaussians $\mathcal{G}^t$ and $\mathcal{G}^{t+1}$, respectively. A sensitivity reward $R^t$ is then derived by comparing the score of $\mathcal{G}^t$ with that of its offspring in $\mathcal{G}^{t+1}$. Given this reward, established RL algorithms, such as PPO, can be leveraged to optimize the policy network $\mathcal{F}_\theta$.

### 3.3. Sensitivity-based Reward Function

Designing an effective reward function for densification and pruning presents two primary challenges: (1) *delayed rewards*, where the benefits of structural changes only emerge after subsequent optimization; and (2) *spatial coupling*, where $\alpha$-blending obscures the contribution of individual overlapping Gaussians. To address these issues, we propose a sensitivity-based reward function inspired by occlusion sensitivity analysis in Convolutional Neural Networks

(CNNs) (Zeiler & Fergus, 2013).

**Sensitivity Score.** We introduce a sensitivity score, denoted as $\text{Sen}_i$, to quantify the contribution of each Gaussian $\mathcal{G}_i$ to the final reconstruction. Analogous to identifying critical features in CNNs by masking input regions, this score measures the fluctuation in rendering error induced by removing $\mathcal{G}_i$. Intuitively, if removing $\mathcal{G}_i$ significantly degrades rendering quality, the Gaussian is deemed critical and assigned a high score. Conversely, Gaussians that have a negligible impact or introduce artifacts receive low or negative scores. Crucially, this metric disentangles individual contributions from the coupled rendering process and allows for the parallel evaluation of all Gaussians.

Formally, we define the sensitivity score $\text{Sen}_i$ for Gaussian $\mathcal{G}_i$ as the difference in reconstruction error resulting from masking this Gaussian during the rendering process:

$$\text{Sen}_i = \sum_{p \in \text{pix}(\mathcal{G}_i)} \Big( |C_{-i}(p) - C_{gt}(p)| - |C(p) - C_{gt}(p)| \Big),$$
(5)

where $C(p)$ and $C_{-i}(p)$ denote the rendered colors of pixel $p$ using the full set of Gaussians and the subset excluding $\mathcal{G}_i$, respectively. $C_{gt}(p)$ is the ground truth color, and $\text{pix}(\mathcal{G}_i)$ represents the set of pixels covered by the 2D footprint of $\mathcal{G}_i$. An illustration of the sensitivity score is provided in the bottom-left of Figure 2.

**Sensitivity Reward.** We calculate the reward $\mathcal{R}_i^t$ for each action in $\mathcal{A}^t$ by measuring the sensitivity gain from the parent Gaussians in $\mathcal{G}^t$ to their offspring in $\mathcal{G}^{t+1}$. A mapping is maintained to track this correspondence. We define $\mathcal{R}_i^t$ as

the net sensitivity gain yielded by the offspring of $\mathcal{G}_i^t$.

$$\mathcal{R}_i^t = \left( \sum_{j \in \text{child}(\mathcal{G}_i^t)} Sen_j^{t+1} \right) - Sen_i^t, \qquad (6)$$

where $\text{child}(\mathcal{G}_i^t)$ represents the set of offspring Gaussians in $\mathcal{G}^{t+1}$ derived from $\mathcal{G}_i^t$ via action $\mathcal{A}_i^t$. Note that for pruned Gaussians, the aggregate sensitivity of their offspring (an empty set) is defined as zero. An illustration of our sensitivity reward is provided in the bottom-right of Figure 2.

### 3.4. Efficient-yet-Exact Formulation of Sensitivity Score

Naively computing the sensitivity score requires re-rendering the scene for each Gaussian to evaluate $C_{-i}(p)$, leading to a prohibitive $O(N^2)$ complexity. To ensure efficiency, we derive a closed-form solution based on the $\alpha$-blending formulation. By reusing the accumulated transmittance and final color from the forward pass, we can compute $C_{-i}(p)$ analytically without re-rendering.

The standard $\alpha$-blending equation for a pixel $p$ with $N$ Gaussians (sorted by depth) is:

$$C(p) = \sum_{k=1}^{N} T_k \alpha_k c_k. \qquad (7)$$

Here, $T_k$ represents the accumulated transmittance, defined as $T_k = \prod_{j=1}^{k-1}(1 - \alpha_j)$, with $T_1 = 1$. When removing Gaussian $\mathcal{G}_i$, the rendered color without Gaussian $\mathcal{G}_i$ can be expressed as:

$$C_{-i}(p) = \sum_{k=1}^{i-1} T_k \alpha_k c_k + \sum_{k=i+1}^{N} \left( \prod_{j=1, j\neq i}^{k-1} (1 - \alpha_j) \right) \alpha_k c_k$$

$$= \sum_{k=1}^{i-1} T_k \alpha_k c_k + \frac{1}{1 - \alpha_i} \sum_{k=i+1}^{N} T_k \alpha_k c_k. \qquad (8)$$

Let $\Sigma_k \triangleq \sum_{j=1}^{k} T_j \alpha_j c_j$ represent the accumulated color up to the $k$-th Gaussian. We can rewrite the components of $C_{-i}(p)$ using $\Sigma_k$ and $C(p)$:

$$\sum_{k=1}^{i-1} T_k \alpha_k c_k = \Sigma_{i-1} \qquad (9)$$

$$\sum_{k=i+1}^{N} T_k \alpha_k c_k = C(p) - \Sigma_i \qquad (10)$$

Substituting these back into (8), we obtain the efficient formulation for $C_{-i}(p)$:

$$C_{-i}(p) = \Sigma_{i-1} + \frac{C(p) - \Sigma_i}{1 - \alpha_i}. \qquad (11)$$

This formulation reduces the complexity to $O(N)$, requiring only two linear passes over the sorted Gaussians (one for $C(p)$ and $\Sigma_i$, and another for $C_{-i}(p)$). Given that density control occurs periodically (e.g., every 100 iterations), the computational overhead is negligible (see runtime analysis in Figure 5).

### 3.5. Training

A key component of our approach is the training methodology. We adopt Proximal Policy Optimization (PPO) (Schulman et al., 2017) to optimize our policy network $\mathcal{F}_\theta$ and propose an alternative advantage estimator that avoids learning a separate critic network.

In PPO, the policy network $\mathcal{F}_\theta$ is updated using an advantage estimate $\hat{A}_t$, which indicates whether action $a_t$ performs better or worse than the baseline at state $s_t$. This advantage provides the optimization direction for increasing or decreasing the likelihood of the sampled action. PPO then measures the policy change through the probability ratio between the current and old policies, defined as:

$$r_t(\theta) = \mathcal{F}_\theta(a_t \mid s_t)/\mathcal{F}_{\theta_{\text{old}}}(a_t \mid s_t). \qquad (12)$$

The PPO objective function, with a clipping mechanism, is commonly formulated as:

$$J^{\text{CLIP}}(\theta) = \mathbb{E}_t \Big[ \min \Big( r_t(\theta)\hat{A}_t, \\ \text{clip}\big(r_t(\theta), 1 - \epsilon, 1 + \epsilon\big)\hat{A}_t \Big) \Big]. \qquad (13)$$

A prevalent choice for $\hat{A}_t$ is Generalized Advantage Estimation (GAE) (Schulman et al., 2015b):

$$\hat{A}_t^{\text{GAE}(\gamma, \lambda)} = \sum_{l=0}^{\infty} (\gamma\lambda)^l \, \delta_{t+l}^V, \qquad (14)$$

where $\delta_t^V$ represents the temporal difference (TD) error:

$$\delta_t^V = r_t + \gamma V_\phi(s_{t+1}) - V_\phi(s_t). \qquad (15)$$

Here, $V_\phi$ denotes the critic network, and $\gamma$ and $\lambda$ are hyperparameters. Given the sensitivity reward proposed in Section 3.3, we train the policy network via PPO to achieve density control. Although PPO is generally robust, its direct application in our setting is challenging because advantage estimation critically relies on a well-trained critic $V_\phi$. In our problem, the reward signal is induced by operations over a very large set of Gaussians, making accurate value prediction difficult, especially early in training. Consequently, inaccurate value estimates can dominate the advantage computation, hindering optimization and potentially leading to excessive redundant densification.

*Table 1.* Quantitative comparison of LeGS with state-of-the-art methods on novel view synthesis. We report image quality metrics: PSNR (↑), SSIM (↑), and LPIPS (↓), where ↑ indicates higher is better and ↓ indicates lower is better. For efficiency, we report the final number of Gaussians (#G, in millions, ↓). The best and second-best results for each column are highlighted. FastGS* is a variant of FastGS where the gradient threshold is adjusted to match the Gaussian count of our method (LeGS) for a fair comparison.

| Method | Mip-NeRF 360 | | | | Tanks & Temples | | | | Deep Blending | | | |
|---|---|---|---|---|---|---|---|---|---|---|---|---|
| | SSIM↑ | PSNR↑ | LPIPS↓ | #G↓ | SSIM↑ | PSNR↑ | LPIPS↓ | #G↓ | SSIM↑ | PSNR↑ | LPIPS↓ | #G↓ |
| 3DGS | 0.815 | 27.59 | 0.215 | 2.74 | 0.854 | 23.75 | 0.169 | 1.68 | 0.907 | 29.80 | 0.238 | 2.48 |
| Pixel-GS | 0.824 | 27.56 | 0.190 | 5.59 | 0.857 | 23.80 | 0.149 | 4.52 | 0.896 | 28.97 | 0.248 | 4.65 |
| Taming-3DGS | 0.829 | 27.85 | 0.208 | 3.22 | 0.856 | 24.17 | 0.168 | 1.84 | 0.909 | 29.98 | 0.234 | 2.80 |
| 3DGS-MCMC | 0.831 | 27.81 | 0.192 | 3.38 | 0.862 | 24.23 | 0.155 | 1.85 | 0.903 | 29.57 | 0.243 | 2.95 |
| Perceptual-GS | 0.829 | 27.71 | 0.189 | 2.69 | 0.856 | 23.71 | 0.152 | 1.72 | 0.906 | 29.83 | 0.232 | 2.89 |
| FastGS | 0.820 | 27.87 | 0.216 | 1.03 | 0.858 | 24.49 | 0.174 | 0.55 | 0.911 | 30.17 | 0.240 | 0.65 |
| FastGS* | 0.824 | 27.73 | 0.186 | 2.49 | 0.864 | 24.21 | 0.146 | 1.79 | 0.909 | 30.00 | 0.236 | 1.12 |
| LeGS | 0.837 | 28.30 | 0.184 | 2.17 | 0.871 | 24.74 | 0.142 | 1.63 | 0.914 | 30.35 | 0.227 | 1.05 |

*Table 2.* Time breakdown (in seconds) across different datasets and methods.

| Dataset | Method | Total | Render | Densify&Prune | Reward | RL Training |
|---|---|---|---|---|---|---|
| MipNeRF360 | 3DGS-MCMC | 2639 | 383 | 6 | – | – |
| | FastGS* | 599 | 112 | 42 | – | – |
| | LeGS | 745 | 128 | 58 | 46 | 68 |
| Tanks&Temples | 3DGS-MCMC | 1385 | 231 | 4 | – | – |
| | FastGS* | 425 | 84 | 29 | – | – |
| | LeGS | 549 | 103 | 43 | 36 | 58 |
| DeepBlending | 3DGS-MCMC | 2244 | 276 | 5 | – | – |
| | FastGS* | 377 | 61 | 30 | – | – |
| | LeGS | 453 | 79 | 38 | 28 | 50 |

*Table 3.* Peak GPU memory usage (GB)

| Method | MipNeRF360 | Tanks & Temples | Deep Blending |
|---|---|---|---|
| 3DGS-MCMC | 11.94 | 6.73 | 9.47 |
| FastGS* | 8.99 | 7.80 | 6.01 |
| LeGS | 12.57 | 9.6 | 8.12 |

**Maintain Action-Based Value.** To mitigate this issue, inspired by (Shao et al., 2024), we propose replacing the critic value with an average value derived from the *maintain* action. In PPO, the critic's value function primarily serves as a baseline for the advantage estimate. Since the *maintain* action corresponds to performing no operation, it naturally serves as a state-dependent baseline: densification should only be encouraged when it yields a higher reward than doing nothing. Specifically, we use the average reward of the *maintain* action at time step $t$, denoted by $\bar{r}(maintain, t)$, to form a GAE-style estimator:

$$\hat{A}_t^{\text{GAE}(\gamma, \lambda)} = \sum_{l=0}^{\infty} (\gamma\lambda)^l \delta_{t+l}^{\bar{r}}, \quad (16)$$

$$\delta_t^{\bar{r}} = r_t + \gamma\, \bar{r}(maintain, t+1) - \bar{r}(maintain, t). \quad (17)$$

**Gaussian Optimization.** To train the learnable parameters of the Gaussians, we follow standard 3DGS procedures.

We optimize these parameters with respect to an $\mathcal{L}_1$ loss on rendered pixel colors, combined with an SSIM term (Wang et al., 2004) $\mathcal{L}_{\text{SSIM}}$:

$$\mathcal{L} = (1 - \lambda_l) \times \mathcal{L}_1 + \lambda_l \times (1 - \mathcal{L}_{\text{SSIM}}), \quad (18)$$

where $\lambda_l$ is a hyperparameter.

**Optimality and convergence.** Our analysis focuses on one-step local optimality rather than global optimality. We show that, with bounded surrogate error $\delta$, maximizing the proposed Sensitivity Reward yields an action whose exact one-step gain is at most $2\delta$-suboptimal. Moreover, with a positive acceptance margin larger than the global reward error, the image-level reconstruction loss decreases monotonically and the editing process terminates in finitely many steps. The full proof is provided in Appendix B.

## 4. Experiments

### 4.1. Experimental Setup

**Datasets and metrics.** Following the experimental protocol of 3DGS (Kerbl et al., 2023), we conduct experiments on three representative real-world datasets widely used for novel view synthesis evaluation: Mip-NeRF 360 (Barron et al., 2022), Deep-Blending (Hedman et al., 2018), and

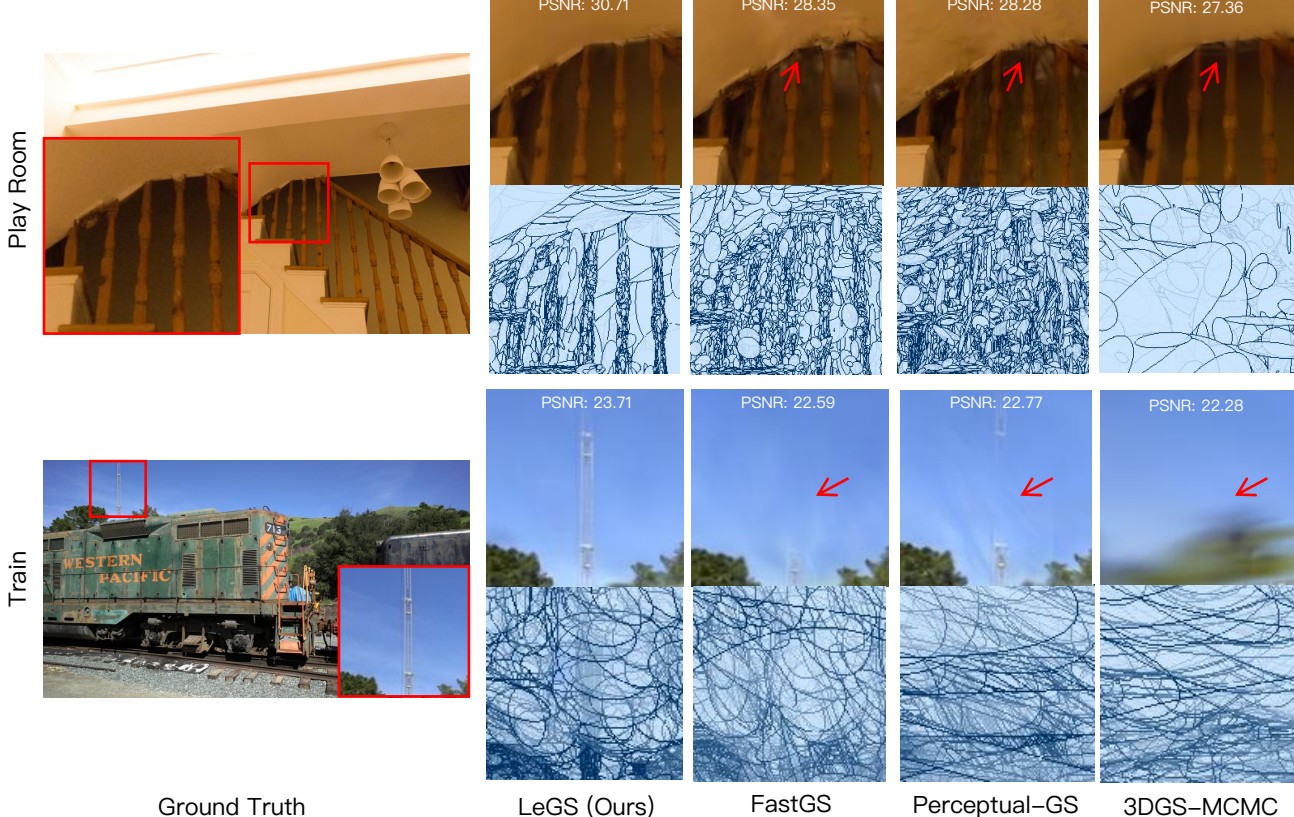

*Figure 3.* Qualitative comparison of our LeGS against state-of-the-art methods on the *Play Room*, and *Train*. The first row of each scene shows the rendering results, while the second row shows the visualization of the Gaussians.

Tanks & Temples (Knapitsch et al., 2017). To evaluate performance, we report common quality metrics: Peak Signal-to-Noise Ratio (PSNR), Structural Similarity (SSIM) (Wang et al., 2004), and Perceptual Similarity (LPIPS) (Zhang et al., 2018). We present the final number of Gaussians (#G) as the efficiency metric.

**Baseline.** We select methods that enhance 3DGS performance through optimized density control strategies, similar to our approach, for comparison to validate the effectiveness of our proposed method. Specifically, we compare against state-of-the-art methods, including PixelGS (Zhang et al., 2024), Taming-3DGS (Mallick et al., 2024), 3DGS-MCMC (Kheradmand et al., 2024), Perceptual-GS (Zhou & Ni, 2025), FastGS (Ren et al., 2025), and the vanilla 3DGS (Kerbl et al., 2023) as baselines.

**Implementation Details.** Our method is built upon the FastGS (Ren et al., 2025) framework, leveraging its efficient rendering capabilities. It is important to note that while we adopt the rendering backend of FastGS, our proposed learning-based paradigm is fundamentally distinct from the heuristic density control employed in the original FastGS. All experiments are conducted on a single NVIDIA L20 GPU. To ensure a fair comparison, all baseline methods

are implemented using their official codebases and adhere to the standard evaluation protocols of vanilla 3DGS. In particular, because COLMAP image settings and the image resolutions in Mip-NeRF 360 differ across prior works, we consistently follow the standard configuration from the official 3DGS code repository for all experiments. Additional implementation details are provided in Section A.

### 4.2. Comparisons with State-of-the-Art

**Quantitative Results.** Table 1 presents a quantitative comparison of LeGS against state-of-the-art methods in terms of novel view synthesis quality. For a fair comparison, we introduce FastGS*, a baseline variant where the gradient threshold is adjusted to match its Gaussian count with that of our method. Surprisingly, despite employing a higher number of Gaussians, FastGS* does not consistently outperform the original FastGS and sometimes exhibits degradation in SSIM and PSNR metrics. This can be attributed to FastGS's complex heuristic rules, which resulted in insufficient robustness across varying quantities of Gaussians. Further experiments on the relationship between rendering quality and Gaussian count are detailed in Figure 4.

Across all three datasets, our method consistently achieves

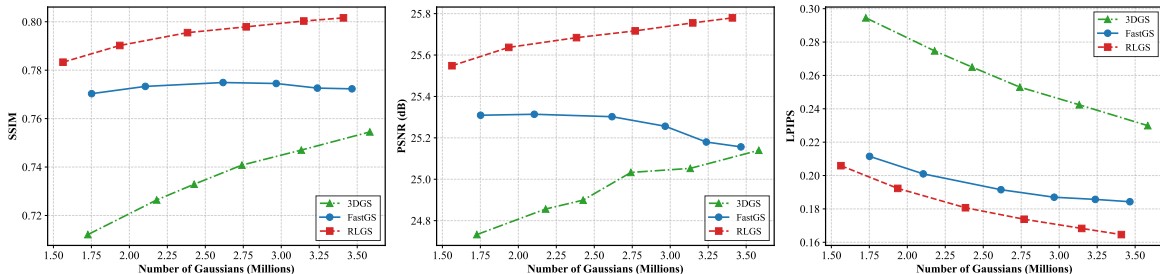

*Figure 4.* Relationship between reconstruction quality and the number of Gaussians across different methods. We compare vanilla 3DGS, FastGS, and our method, LeGS, under varying numbers of Gaussians for novel view synthesis. LeGS consistently outperforms 3DGS and FastGS across all three metrics (SSIM↑, PSNR↑, LPIPS↓), demonstrating superior adaptability and robustness in resource allocation. Notably, FastGS exhibits degradation in SSIM and PSNR when the number of Gaussians increases beyond a certain point, highlighting the limitations of delicate heuristic rules in maintaining robustness across varying scenarios.

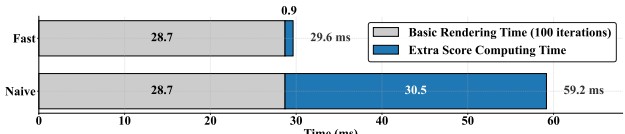

*Figure 5.* Efficiency evaluation of our proposed fast sensitivity score. "Naive" refers to a baseline approach that computes the score by re-rendering at each step. "Fast" represents our efficient closed-form solution defined in Equation (8).

the best performance, particularly excelling in PSNR and LPIPS metrics. This demonstrates the superiority of our learnable densification strategy. We observe that our method yields the most significant performance gains on Mip-NeRF 360 and Tanks & Temples, which are complex, large-scale scenes demanding a higher density of Gaussians for accurate representation. Specifically, LeGS outperforms 3DGS by **+0.81** dB in PSNR on Mip-NeRF 360 and **+0.99** dB on Tanks & Temples. This aligns with our expectations: larger and more complex scenes necessitate extensive densification, where an intelligent allocation strategy demonstrates its most significant impact. In contrast, for the Deep Blending dataset, which generally requires less aggressive densification, our learnable approach yields relatively modest, though still positive, improvements.

In terms of efficiency, our method maintains a compact Gaussian representation without compromising quality. Specifically, it yields the second-fewest Gaussians on the Mip-NeRF 360 and Tanks & Temples datasets, while achieving the lowest count on Deep Blending. Crucially, we observe a distinct trend in resource allocation: while heuristic methods generally increase Gaussian counts for Tanks & Temples compared to Deep Blending, our learnable approach reveals that Tanks & Temples indeed requires a significantly higher density of Gaussians. This contrast suggests that heuristic criteria can be suboptimal in resource allocation, whereas our learned policy adaptively distributes Gaussians based on actual scene complexity. This behavior

validates that LeGS intelligently identifies regions requiring denser representations, allocating computational resources precisely where they are needed most.

### 4.3. Ablation Studies

**Qualitative Results.** Figure 3 presents visual comparisons of novel view synthesis on the *Train*, and *Play Room* scenes against three SOTA baselines: FastGS, Perceptual-GS, and 3DGS-MCMC. The second row of each scene shows the visualization of the Gaussians. These results demonstrate that LeGS achieves superior visual quality by learning Gaussians whose distributions are better aligned with object geometry and texture across diverse scenes, validating our learnable density control paradigm.

**Efficiency Analysis.** As shown in Table 2, LeGS achieves training speeds comparable to FastGS, effectively mitigating the overhead introduced by the neural network and sensitivity score computation. Compared to FastGS*, LeGS incurs an additional 80–100 seconds due to reward computation and RL training.

Notably, LeGS maintains densification and pruning times on par with FastGS, despite incorporating sensitivity scoring and RL forward passes, underscoring the efficiency of its closed-form sensitivity formulation.

Although our method introduces some additional GPU memory and computational overhead, the overall resource usage remains comparable to prior work.

**Relationship between reconstruction quality and Gaussian quantity.** We investigate the reconstruction quality as a function of the number of Gaussians for vanilla 3DGS, FastGS, and our proposed LeGS. As depicted in Figure 4, FastGS exhibits sensitivity to the quantity of Gaussians, resulting in performance degradation with increasing Gaussian count. In contrast, our learnable paradigm, LeGS, demonstrates robust scalability, consistently improving re-

*Table 4.* The proportion of locally optimal actions for LeGS and FastGS across different training iterations.

|        | 3000   | 5000   | 7000   | 9000   | 11000  | 13000  | 15000  |
|--------|--------|--------|--------|--------|--------|--------|--------|
| FastGS | 26.39% | 29.17% | 33.77% | 33.95% | 35.25% | 36.22% | 35.60% |
| LeGS   | 49.34% | 53.43% | 59.01% | 62.36% | 63.62% | 64.09% | 64.85% |

*Table 5.* Ablation study on the effectiveness of our proposed learnable densification (LD) and pruning strategies (LP) on the Mip-NeRF 360 dataset. FastGS* serves as a baseline, where its gradient threshold is tuned to achieve a Gaussian count comparable to our full model.

| Method      | Mip-NeRF 360 |        |         |       |
|-------------|--------------|--------|---------|-------|
|             | SSIM↑        | PSNR↑  | LPIPS↓  | #G↓   |
| FastGS*     | 0.824        | 27.73  | 0.186   | 2.49  |
| + LD        | 0.831        | 28.14  | 0.183   | 2.48  |
| + LP (FULL) | 0.837        | 28.30  | 0.184   | 2.17  |

*Table 6.* Ablation study on the core components of LeGS: the proposed *maintain* Action-based Value (KAV) and the Sensitivity-based Reward (SR) function, evaluated on the Mip-NeRF 360.

| Method      | Mip-NeRF 360 |        |         |       |
|-------------|--------------|--------|---------|-------|
|             | SSIM↑        | PSNR↑  | LPIPS↓  | #G↓   |
| w/o SR      | 0.826        | 28.01  | 0.191   | 2.12  |
| w/o KAV     | 0.832        | 28.17  | 0.181   | 2.63  |
| LeGS (FULL) | 0.837        | 28.30  | 0.184   | 2.17  |

construction quality across a wide range of Gaussian counts. These findings underscore the robustness of our learning-based density control paradigm.

**Effect of learnable strategy.** As presented in Table 5, we evaluate the individual contributions of Learnable Densification (LD) and Learnable Pruning (LP) on the Mip-NeRF 360 dataset. To ensure a fair comparison, we employ FastGS* as the baseline. The results demonstrate that integrating our Learnable Densification (+LD) yields a significant performance boost, improving PSNR from 27.73 dB to 28.14 dB. Furthermore, incorporating Learnable Pruning (+LP) effectively removes redundant Gaussians while further enhancing performance. Notably, this pruning step not only improves compactness but also further enhances reconstruction quality (28.30 dB PSNR), indicating that our strategy effectively eliminates noisy or unnecessary Gaussians.

**Effect of proposed component.** We evaluate the individual impact of our proposed Sensitivity-based Reward (SR) function and the *maintain* Action-based Value (KAV) estimation on the Mip-NeRF 360 dataset. For the "w/o SR" ablation, we replace our SR function with a baseline reward that attributes per-pixel reconstruction improvements to individual Gaussians solely based on their rendering weights. As shown in Table 6, this variant exhibits performance degra-

dation. This can be attributed to the baseline reward's inability to accurately disentangle individual Gaussian contributions from the complex blending process, leading to imprecise credit assignment. The "w/o KAV" variant substitutes our proposed *maintain* Action-based Value estimation with a standard Proximal Policy Optimization (PPO) critic network for value prediction. While this configuration achieves performance comparable to our full model, it leads to a substantial increase in redundant Gaussians, with the Gaussian count rising from 2.17M to 2.63M. This highlights KAV's better guidance for training.

**Efficiency of Fast Sensitivity Score.** We evaluate the computational cost of the naive calculation of the sensitivity score (Naive) against our efficient closed-form solution (Fast). Runtime measurements were conducted on an NVIDIA L20 GPU using the *Bicycle* scene from the Mip-NeRF 360 dataset with a workload of 100 Gaussians per tile. As presented in Figure 5, the Naive approach is computationally prohibitive due to its $O(N^2)$ complexity, requiring 30.5 ms. In contrast, our optimized calculation achieves linear $O(N)$ complexity, dramatically reducing the computation time to 0.9 ms, which incurs a negligible extra computing time beyond the basic rendering time.

**Local optimality.** As shown in Table 4, we compare the actions predicted by LeGS and FastGS against defined "locally optimal actions" across iterations. Specifically, we sample 5% of Gaussians, evaluate all four actions {maintain, clone, split, prune}, and select the one yielding the largest PSNR improvement after 50 iterations as the local optimum. LeGS matches these locally optimal actions more frequently than FastGS, demonstrating the effectiveness of learnable RL-based density control over heuristic strategies.

## 5. Conclusion

This paper presents LeGS, a framework that transitions adaptive density control from heuristic-based rules to a learning paradigm. We propose a fast sensitivity-based reward function that quantifies each primitive's contribution to rendering quality. Furthermore, we stabilize training by *maintain*-action value estimation. Extensive experiments demonstrate that LeGS achieves SOTA rendering quality with compact representations. We believe this work establishes a foundation for learning-based density control in 3DGS.

## Impact Statement

This paper presents work whose goal is to advance the field of Machine Learning. There are many potential societal consequences of our work, none which we feel must be specifically highlighted here.

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

# A. Additional Implementation Details

**Calculation of Gradient and Sensitivity Score.**    During training, we randomly sample $K$ views from the training set to render and compute the gradients of Gaussian attributes and the sensitivity score. Specifically, we render these views, then calculate the loss with respect to the ground truth using Equation (18), and subsequently perform backpropagation to obtain the gradients of Gaussian attributes. The ground truth is also used to calculate the sensitivity score according to Equation (5). Following the CUDA parallelism employed in the vanilla rendering process, we split each image into tiles and perform calculations for each tile in parallel. For sorted Gaussians within a tile, we first traverse the Gaussians to compute the full rendering result. Subsequently, we traverse the Gaussians a second time, recording the current and previously accumulated color to calculate the rendering result without each individual Gaussian, as described by Equation (8). Finally, we calculate the sensitivity score using Equation (5). Note that, we calculate the sensitivity score at time steps $t$ and $t+1$ using the same set of $K$ sampled views. Practically, we find that setting $K$ to 10 is sufficient to cover most Gaussians in the scene.

**Pseudo Code.**    The pseudo code for our framework is presented in Algorithm 1. In this algorithm, vanilla density control operations are denoted in gray, whereas our novel learning-based paradigm is highlighted in blue. Specifically, we parameterize the density control criteria, employ a neural network to determine the appropriate action for each Gaussian, and optimize this network using our proposed sensitivity reward function via reinforcement learning.

---

*Algorithm 1.* Comparison our learnable density control with vanilla

---

$M \leftarrow$ SfM Points {Positions}
$S, C, A \leftarrow$ InitAttributes() {Covariances, Colors, Opacities}
$i \leftarrow 0$ {Iteration Count}
$\mathcal{F}_\theta \leftarrow$ InitNetwork()
**while** not converged **do**
   $V, \hat{I} \leftarrow$ SampleTrainingView() {Camera $V$ and Image}
   $I \leftarrow$ Rasterize($M, S, C, A, V$)
   $L \leftarrow Loss(I, \hat{I})$ {Loss}
   $M, S, C, A \leftarrow$ Adam($\nabla L$) {Backprop & Step}
   **if** IsRefinementIteration($i$) **then**
      **for all** Gaussians $(\mu, \Sigma, c, \alpha)$ **in** $(M, S, C, A)$ **do**
         {Vanilla Density Control}
         **if** $\alpha < \epsilon$ OR IsTooLarge($\mu, \Sigma$) **then**
            RemoveGaussian()
         **end if**
         **if** $\nabla_p L > \tau_p$ **then**
            **if** $\|S\| > \tau_S$ **then**
               SplitGaussian($\mu, \Sigma, c, \alpha$)
            **else**
               CloneGaussian($\mu, \Sigma, c, \alpha$)
            **end if**
         **end if**
         {Our Learnable Density Control}
         $Sen \leftarrow$ CalculateSensitivityScore($\mu, \Sigma, c, \alpha, \hat{I}$)
         $\mathcal{A} \leftarrow \mathcal{F}_\theta(\mu, \Sigma, c, \alpha, Sen)$
         ExecuteAction($\mu, \Sigma, c, \alpha, \mathcal{A}$)
         $Sen' \leftarrow$ CalculateSensitivityScore($\mu, \Sigma, c, \alpha, \hat{I}$)
         $R \leftarrow$ CalculateReward($Sen, Sen'$)
         $\mathcal{F}_\theta \leftarrow$ UpdateNetwork($\mathcal{F}_\theta, R$)
      **end for**
   **end if**
   $i \leftarrow i + 1$
**end while**

---

**Fast Score derivation.** The details of the fast score derivation are as follows:

$$C(p) = \sum_{i=1}^{N} T_i \alpha_i c_i, \quad \text{where} \quad T_i = \prod_{j=1}^{i-1} (1 - \alpha_j) \tag{19}$$

$$
\begin{aligned}
C_{-i}(p) &= \sum_{j=1}^{i-1} T_j \alpha_j c_j + \sum_{j=i+1}^{N} \frac{T_j}{1 - \alpha_i} \alpha_j c_j \\
&= \sum_{j=1}^{i-1} T_j \alpha_j c_j + \frac{1}{1 - \alpha_i} \sum_{j=i+1}^{N} T_j \alpha_j c_j \\
&= \Sigma_{i-1} + \frac{\Sigma_N - \Sigma_i}{1 - \alpha_i}, \quad \text{where} \quad \Sigma_i \triangleq \sum_{j=1}^{i} T_j \alpha_j c_j \\
&= \Sigma_{i-1} + \frac{C(p) - \Sigma_i}{1 - \alpha_i} \tag{20}
\end{aligned}
$$

**Details of Policy Network.** We employ a 3-layer MLP with SwiGLU activation functions as an encoder to process the input. This encoder is subsequently followed by two distinct heads responsible for predicting action probabilities. Specifically, a "Densification Head" predicts the probabilities for *maintain*, *clone*, and *split* actions, while a "Pruning Head" predicts the probability of a *prune* action. The hidden dimension for all layers within the MLP is set to 64.

**Hyperparameters** . The training hyperparameters used for PPO are listed in Table 7. The reward is computed at the same frequency as densification, every 100 iterations.

*Table 7.* Hyperparameter settings.

| Discount Factor | GAE Parameter | Policy Clip Ratio | Optimization Epochs | n-step TD | Initial LR | Final LR |
|---|---|---|---|---|---|---|
| 0.99 | 0.95 | 0.2 | 2 | 2 | 1e−3 | 1e−5 |

# B. Proof of Optimality and Convergence.

Prior RL theory usually guarantees **monotonic improvement across policy updates**, rather than global optimality (Schulman et al., 2015a). For instance, TRPO provides a monotonic improvement lower bound:

$$\eta(\tilde{\pi}) \geq L_{\pi}(\tilde{\pi}) - \frac{4\epsilon\gamma}{(1 - \gamma)^2} \alpha^2,$$

implying that **one-step improvement** can be guaranteed via a surrogate lower bound (**?**).

Consistent with this perspective, below we provide an approximate bound on the deviation of our one-step decision from the global one-step optimum.

Recall the Sensitivity Score in our paper, which measures each Gaussian's contribution at time step $t$:

$$\text{Sen}_i^t = \sum_{p \in \text{pix}(\mathcal{G}_i^t)} \left( \left| C_{-i}^t(p) - C_{\text{gt}}(p) \right| - \left| C^t(p) - C_{\text{gt}}(p) \right| \right),$$

where $C^t(p)$ and $C_{-i}^t(p)$ are the rendered colors using all Gaussians and excluding $\mathcal{G}_i^t$, respectively. $C_{\text{gt}}(p)$ is the ground-truth color, and $\text{pix}(\mathcal{G}_i^t)$ denotes pixels covered by the 2D footprint of $\mathcal{G}_i^t$.

We also define the Sensitivity Reward, which measures the sensitivity gain from a parent Gaussian to its offspring after action $a$:

$$R_i^t = \left( \sum_{j \in \text{child}(\mathcal{G}_i^t)} \text{Sen}_j^{t+1} \right) - \text{Sen}_i^t.$$

Based on these definitions, we define the reconstruction loss for pixel $p$ as an optimality measure:

$$\ell_p(C) = |C - C_{\text{gt}}(p)|.$$

Maximizing the Sensitivity Reward is approximately equivalent to maximizing one-step optimality, as follows.

For action $a$, let the rendered color be $C_a^{t+1}(p)$ after applying $a$ to $\mathcal{G}_i^t$, and let the rendering after removing the offspring group be $B(p)$. Since $a$ only replaces the parent with its offspring, we have

$$B(p) = C_{-i}^t(p).$$

Define the offspring-group sensitivity similarly to $\text{Sen}_i^t$:

$$\text{Sen}_{\text{child},p}^{t+1}(a) = \ell_p(B(p)) - \ell_p(C_a^{t+1}(p)).$$

Then the exact sensitivity gain from the parent Gaussian to its offspring group can be computed as

$$R_{i,p}^{\star}(a) = \text{Sen}_{\text{child},p}^{t+1}(a) - \text{Sen}_{i,p}^t$$
$$= \ell_p(C^t(p)) - \ell_p(C_a^{t+1}(p)).$$

Since $R_{i,p}^{\star}(a)$ is determined solely by $\ell_p(C_a^{t+1}(p))$, maximizing $R_{i,p}^{\star}(a)$ is exactly equivalent to minimizing $\ell_p(C_a^{t+1}(p))$. **Hence, maximizing the exact sensitivity gain directly yields a one-step optimum.**

Accordingly, our Sensitivity Reward $R_i^t$ serves as a tractable surrogate for the exact gain $R_{i,p}^{\star}(a)$:

$$R_{i,p}(a) = \sum_{j \in \text{child}(\mathcal{G}_i^t)} \text{Sen}_{j,p}^{t+1} - \text{Sen}_{i,p}^t = R_{i,p}^{\star}(a) + \varepsilon_{i,p}(a),$$

where $\varepsilon_{i,p}(a)$ is the interaction error induced by $\alpha$-blending, **which is zero when offspring contributions are pixel-wise non-interacting and small when interactions are weak.**

**One-step local approximate optimality.** Assume that

$$|\varepsilon_{i,p}(a)| \leq \delta$$

for all $a$, and let

$$a^{\star} = \arg\max_a R_{i,p}^{\star}(a), \qquad \hat{a} = \arg\max_a R_{i,p}(a).$$

Then the following properties hold.

**1. Exact recovery under a margin condition.** If

$$R_{i,p}^{\star}(a^{\star}) - \max_{a \neq a^{\star}} R_{i,p}^{\star}(a) > 2\delta,$$

then

$$\hat{a} = a^{\star}.$$

**2. Approximate optimality without a margin condition.** We have

$$R_{i,p}^{\star}(a^{\star}) - R_{i,p}^{\star}(\hat{a}) \leq 2\delta.$$

Equivalently,

$$\ell_p(C_{\hat{a}}^{t+1}(p)) \leq \ell_p(C_{a^{\star}}^{t+1}(p)) + 2\delta.$$

Thus, our chosen action is at most $2\delta$-suboptimal for the **one-step local** decision at the pixel level.

**Proof.** Since $\hat{a}$ maximizes the surrogate reward $R_{i,p}$, we have

$$R_{i,p}(\hat{a}) \geq R_{i,p}(a^{\star}).$$

Using

$$R_{i,p}(a) = R_{i,p}^{\star}(a) + \varepsilon_{i,p}(a),$$

we obtain

$$R_{i,p}^{\star}(\hat{a}) + \varepsilon_{i,p}(\hat{a}) \geq R_{i,p}^{\star}(a^{\star}) + \varepsilon_{i,p}(a^{\star}).$$

Therefore,

$$R_{i,p}^{\star}(a^{\star}) - R_{i,p}^{\star}(\hat{a}) \leq \varepsilon_{i,p}(\hat{a}) - \varepsilon_{i,p}(a^{\star}).$$

Since $|\varepsilon_{i,p}(a)| \leq \delta$, we have

$$\varepsilon_{i,p}(\hat{a}) - \varepsilon_{i,p}(a^{\star}) \leq 2\delta.$$

Thus,

$$R_{i,p}^{\star}(a^{\star}) - R_{i,p}^{\star}(\hat{a}) \leq 2\delta.$$

Moreover, if the margin condition

$$R_{i,p}^{\star}(a^{\star}) - \max_{a \neq a^{\star}} R_{i,p}^{\star}(a) > 2\delta$$

holds, then any $a \neq a^{\star}$ satisfies

$$R_{i,p}^{\star}(a^{\star}) - R_{i,p}^{\star}(a) > 2\delta.$$

By the bound $|\varepsilon_{i,p}(a)| \leq \delta$, we further have

$$R_{i,p}(a^{\star}) > R_{i,p}(a), \qquad \forall a \neq a^{\star}.$$

Hence,

$$\hat{a} = a^{\star}.$$

Since

$$R_{i,p}^{\star}(a) = \ell_p(C^t(p)) - \ell_p(C_a^{t+1}(p)),$$

the approximate optimality bound is equivalent to

$$\ell_p(C_{\hat{a}}^{t+1}(p)) \leq \ell_p(C_{a^{\star}}^{t+1}(p)) + 2\delta.$$

**Image-level convergence.** Let the reconstruction loss for a whole image be

$$L_t = \sum_{p \in \text{pix}} \ell_p(C^t(p)).$$

The exact global one-step sensitivity gain for an image is

$$R_{i,t}^{\star}(a) = L_t - L_{t+1}.$$

Assume the implemented reward satisfies

$$\left| R_i^t(a) - R_i^{\star,t}(a) \right| \leq \delta_g.$$

If we accept an edit only when

$$R_i^t(\hat{a}_t) \geq \tau,$$

then

$$R_i^{\star,t}(\hat{a}_t) \geq \tau - \delta_g.$$

Therefore, if $\tau > \delta_g$, each accepted edit strictly decreases the global loss:

$$L_{t+1} \leq L_t - (\tau - \delta_g).$$

Since $L_t \geq 0$, the loss over the accepted-edit sequence is monotonically decreasing and bounded below, hence convergent. Moreover, the number of accepted edits is finite:

$$T \leq \frac{L_0}{\tau - \delta_g}.$$

For $\tau = 0$, we only obtain

$$L_{t+1} \leq L_t + \delta_g,$$

which gives a local tolerance guarantee but not monotone convergence by itself.

In summary, we do not claim global optimality. Instead, our reward gives a rigorous **one-step local approximate-optimality** view. With bounded reward error and a positive acceptance margin, the editing process yields **monotone loss decrease** and **finite-step termination**.

## C. More Experiments

**Per-scene performance comparisons** of FastGS versus LeGS are reported in Table 8. In scenes with insufficient training views (Treehill), LeGS shows a minor PSNR drop but substantially improves perceptual quality metrics (SSIM and LPIPS). This suggests that LeGS tends to preserve scene structure and texture fidelity even with sparse data.

*Table 8.* Quantitative comparison between FastGS and LeGS on different scenes. We report PSNR (↑), SSIM (↑), and LPIPS (↓).

| Scene | PSNR ↑ | | SSIM ↑ | | LPIPS ↓ | |
|---|---|---|---|---|---|---|
| | FastGS | LeGS | FastGS | LeGS | FastGS | LeGS |
| bicycle | 25.237 | **25.740** (+0.503) | 0.761 | **0.800** (+0.039) | 0.232 | **0.168** (-0.064) |
| flowers | 21.684 | **22.000** (+0.316) | 0.615 | **0.645** (+0.030) | 0.324 | **0.282** (-0.042) |
| garden | 27.697 | **28.066** (+0.369) | 0.873 | **0.881** (+0.009) | 0.099 | **0.090** (-0.010) |
| stump | 26.968 | **27.376** (+0.408) | 0.783 | **0.811** (+0.028) | 0.227 | **0.176** (-0.051) |
| treehill | **22.846** | 22.797 (-0.049) | 0.629 | **0.658** (+0.029) | 0.365 | **0.272** (-0.093) |
| room | 31.963 | **32.482** (+0.519) | 0.923 | **0.931** (+0.008) | 0.212 | **0.193** (-0.019) |
| counter | 29.476 | **29.844** (+0.368) | 0.911 | **0.919** (+0.008) | 0.193 | **0.179** (-0.014) |
| kitchen | 31.967 | **32.673** (+0.705) | 0.934 | **0.936** (+0.002) | 0.115 | **0.114** (-0.000) |
| bonsai | 32.744 | **33.532** (+0.788) | 0.947 | **0.953** (+0.006) | 0.184 | **0.177** (-0.008) |
| truck | 26.105 | **26.532** (+0.427) | 0.889 | **0.897** (+0.007) | 0.139 | **0.121** (-0.018) |
| train | 22.713 | **22.843** (+0.129) | 0.826 | **0.842** (+0.017) | 0.209 | **0.168** (-0.041) |
| drjohnson | 29.685 | **29.763** (+0.079) | 0.908 | **0.910** (+0.002) | 0.244 | **0.223** (-0.021) |
| playroom | 30.687 | **30.856** (+0.169) | **0.915** | 0.914 (-0.001) | 0.236 | **0.218** (-0.018) |

**More qualitative results.** Figure 6 presents a visual comparison of novel view synthesis on the *Play Room*, *Tree Hill*, *Bonsai*, and *Dr Johnson* scenes, contrasting our approach with FastGS, Perceptual-GS, and 3DGS-MCMC. In the *Play Room* scene, our LeGS method achieves superior reconstruction of illumination and color, whereas alternative approaches exhibit blurring or color deviation. For the *Bonsai* scene, our method accurately reconstructs the intricate details of the piano keys, which appear significantly blurred in baseline results. In the *Treehill* scene, our approach particularly excels at recovering high-frequency texture details. As highlighted in the zoom-in view, LeGS precisely captures the granular structure of the gravel ground. In contrast, FastGS, Perceptual-GS and 3DGS-MCMC suffer from under-densification. The *Dr Johnson* scene further demonstrates LeGS's robustness in indoor environments characterized by complex illumination. Our method not only faithfully reconstructs the intricate structure of the chandelier but also maintains a clean and accurate representation of the surrounding ceiling. Conversely, baseline methods struggle considerably with strong glare, leading to blurred lamp geometry and prominent artifacts on the ceiling.

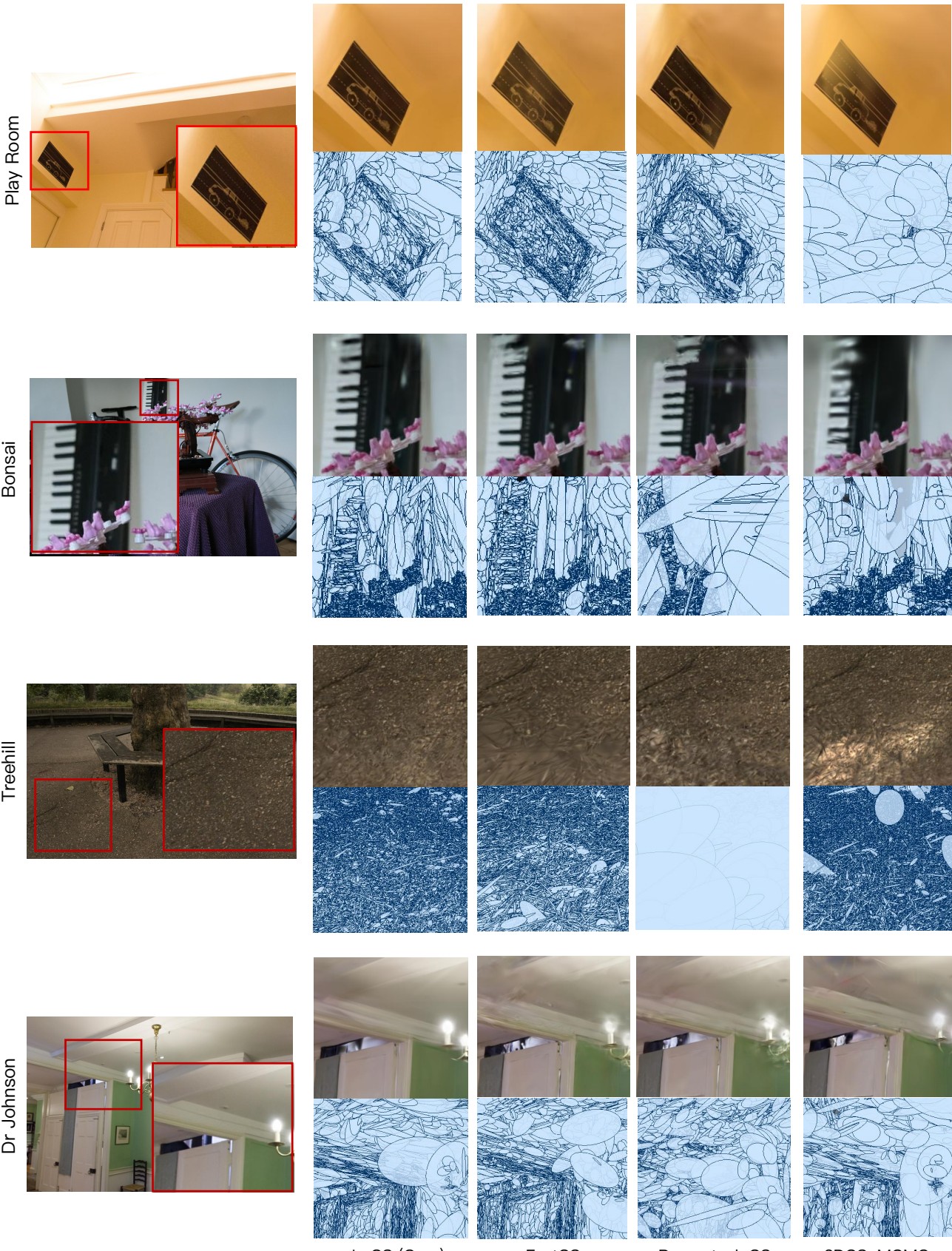

*Figure 6.* More qualitative results on *Play Room*, *Bonsai*, *Tree Hill* and *Dr Johnson*.

