# OpenReview forum: "Beyond Heuristics: Learnable Density Control for 3D Gaussian Splatting"
_ICML.cc/2026/Conference — ICML 2026 regular_

### Official Review · Reviewer_abH7 · 2026-03-09

**Soundness:** 2
**Presentation:** 2
**Significance:** 3
**Originality:** 3
**Overall Recommendation:** 4
**Confidence:** 4

**Summary:**

This paper proposes LeGS, a framework that transitions the adaptive density control in 3D Gaussian splatting (3DGS) from traditional heuristic-based rules to a fully learnable policy. The method formulates density control as a reinforcement learning (RL) problem optimized via PPO, with an action space consisting of maintaining, cloning, splitting, or pruning. To provide effective reward signals, the authors introduce a sensitivity-based reward function that precisely quantifies the marginal contribution of individual Gaussians to the reconstruction quality. Moreover, to overcome the computational bottleneck, the authors derive a closed-form solution for the sensitivity score by reusing the accumulated transmittance from the forward pass, reducing the reward computation complexity from $O(N^2)$ to $O(N)$. Experiments on multiple benchmark datasets demonstrate that the method achieves a better balance between reconstruction quality and the number of Gaussians.

**Compliance With Llm Reviewing Policy:**

Affirmed.

**Final Justification:**

This work proposes learnable density control via RL to address the limitations of the traditional heuristic paradigm in 3DGS, which is interesting. The experiments demonstrate the effectiveness of the proposed method in performance improvements.

The rebuttal addressed my main concerns about the zero-shot generalization capability and the GPU memory consumption introduced by RL. Therefore, I raised my recommendation to 4-Weak Accept.

**Key Questions For Authors:**

1. Have you experimented with training a unified policy network on a set of diverse scenes and applying it zero-shot (with frozen weights) to a novel scene? This would truly prove the method has learned a robust "policy" rather than a per-scene overfitting trick.

2. While Table 4 indicates that the training time of LeGS is comparable to FastGS, there is no discussion regarding the GPU memory footprint. RL algorithms like PPO typically require storing extensive trajectories (e.g., states, actions, rewards, and log probabilities) in memory for advantage estimation and policy updates. Given that the scene contains millions of Gaussians, how does the peak GPU memory (VRAM) consumption of LeGS compare to FastGS during training? Please provide a quantitative comparison.

**Limitations:**

Given the introduction of RL, the authors should explicitly discuss the extreme sensitivity of PPO training to hyperparameters and the potential instability when exploring a massive discrete action space that involves millions of Gaussians. Moreover, while the paper claims that the training time is comparable to FastGS, it fails to discuss the immense GPU memory (VRAM) pressure caused by storing RL trajectory data. A more comprehensive analysis of these limitations and the VRAM overhead should be added to the main text.

**Strengths And Weaknesses:**

**Strengths:**

1. Overcoming the limitations of heuristic density control is a core challenge in the current 3DGS domain. Formulating this as a data-driven Markov decision process (MDP) to dynamically decide whether to maintain, clone, split, or prune is an inspiring and promising perspective.
2. The derivation of $C_{-i}(p)$ from Equation (8) to (11) is a major highlight of this paper. Optimizing the sensitivity calculation, which naively requires quadratic complexity, to a single linear traversal taking only 0.9 ms is a practical technical contribution.
3. The experiments demonstrate significant improvements, particularly in complex large-scale scenes (e.g., Tanks & Temples), where the PSNR increases by +0.99 dB while maintaining a relatively low number of Gaussians.

**Weaknesses:**

1. According to Algorithm 1, the policy network $\mathcal{F}_{\theta}$ is initialized and optimized from scratch for each individual scene alongside the Gaussians. This implies that LeGS does not learn a generalizable density control policy across scenes. Instead, it uses an RL-trained neural network as a highly complex regularization/optimization trick for single-scene fitting. Introducing a PPO training pipeline just to fit a single scene seems heavily over-engineered.
2. Considering the immense system complexity introduced by the policy network and RL training pipeline, the performance gains over the baseline (FastGS) are somewhat limited on certain datasets (e.g., on Mip-NeRF 360 and Deep Blending, the PSNR improvements are roughly around 0.18 to 0.43 dB). It is questionable whether this level of gain sufficiently justifies the inherently high training complexity and debugging costs associated with RL in practical applications.

---

> ### Author Rebuttal · Authors · 2026-03-31
>
> **[Q1] Generalization ability of a unified policy network (zero-shot transfer to novel scenes):**
>
> We thank the reviewer for the valuable suggestion. Following it, we conducted experiments on indoor scenes from the Mip-NeRF 360 dataset. Specifically, we trained the RL network on Counter, Kitchen, and Bonsai, and applied it to Room in a zero-shot manner. The results below show that, even in the zero-shot setting, LeGS still outperforms FastGS without any scene-specific adaptation, demonstrating strong generalization of the learned RL policy. Due to the limited rebuttal time, we trained on only three scenes; we expect that using more training scenes will yield an even more generalizable RL policy and better zero-shot performance, which we will investigate in the final paper.
>
> | Method | **SSIM↑** | **PSNR↑** | **LPIPS↓** |
> | --- | --- | --- | --- |
> | FastGS | 0.923 | 31.96 | 0.212 |
> | LeGS (zero-shot) | 0.927 | 32.14 | 0.199 |
> | LeGS | 0.931 | 32.48 | 0.193 |
>
> **[Q2] Peak GPU memory (VRAM) usage comparison with baseline methods:**
>
> Peak GPU memory usage (GB) is summarized as follows. Although the RL procedure introduces additional intermediate variables and rollout data, the policy network is lightweight, and rollout data are loaded onto the GPU only when needed. As a result, peak GPU memory usage remains comparable to that of baseline methods, with no significant overhead observed in practice.
>
> | Method | MipNerf360 | Tanks & Temples | Deep Blending |
> | --- | --- | --- | --- |
> | 3DGS-MCMC | 11.94 | 6.73 | 9.47 |
> | FastGS* | 8.99 | 7.80 | 6.01 |
> | LeGS | 12.96 | 9.41 | 9.88 |
>
> **Clarification of our contribution and advantage:**
>
> LeGS surpasses FastGS by a margin comparable to that of FastGS over 3DGS, underscoring the significance of our improvements. Despite the additional complexity of RL training and debugging, our method transfers seamlessly from FastGS to 3DGS, demonstrating strong robustness and cross-framework generalization.
>
> | Method | SSIM↑ | PSNR↑ | LPIPS↓ | #G↓ |
> | --- | --- | --- | --- | --- |
> | 3DGS | 0.854 | 23.75 | 0.169 | 1.68 |
> | 3DGS+LeGS | **0.862** | **24.28** | **0.164** | **1.02** |
>
> Our main contribution is reframing density control from heuristic rules to a learnable approach, enabling adaptation across scenes. In response to another reviewer, we evaluate the proportion of RL-predicted actions matching defined “local optimal actions” across iterations and compare it with the FastGS heuristic. Herein, we define “locally optimal action” as follows: we sample 5% of the Gaussians and evaluate all four actions (maintain, clone, split, prune). For each action, we compute the PSNR improvement over the subsequent 50 iterations to identify the “locally optimal action”, and then measure how often the RL policy and FastGS match this choice.
>
> |  | 3000 | 5000 | 7000 | 9000 | 11000 | 13000 | 15000 |
> | --- | --- | --- | --- | --- | --- | --- | --- |
> | FastGS | 26.39% | 29.17% | 33.77% | 33.95% | 35.25% | 36.22% | 35.60% |
> | LeGS | 49.34% | 53.43% | 59.01% | 62.36% | 63.62% | 64.09% | 64.85% |
>
> The results show that LeGS consistently predicts significantly more accurate “locally optimal actions” than FastGS, which empirically demonstrates the superiority of the learnable density control via RL over the heuristic paradigm in terms of convergence, stability, and effectiveness.
>
> While RL does not yet provide a theoretically optimal solution, we present it as an initial but practical step toward more principled, learnable density control. Such a learnable framework opens up opportunities for extensions to dynamic scene reconstruction, and can be naturally integrated with methods such as 4DGS. The following experiments adapt LeGS to 4DGS on the Broom and Banana scenes from HyperNeRF and report PSNR.
>
> |  | Broom | Banana |
> | --- | --- | --- |
> | 4DGS | 22.0 | 28.0 |
> | 4DGS + LeGS | 22.4 | 28.2 |

---

> > ### Author Rebuttal · Reviewer_abH7 · 2026-04-03
> >
> > I thank the authors for their detailed rebuttal and the additional experiments provided.
> >
> > The results on zero-shot generalization and peak GPU memory usage effectively address my concerns. The more accurate “locally optimal actions” also demonstrate the effectiveness of RL. These new results are recommended to be incorporated into the new version.

---

### Official Review · Reviewer_GTjf · 2026-03-09

**Soundness:** 4
**Presentation:** 4
**Significance:** 4
**Originality:** 3
**Overall Recommendation:** 5
**Confidence:** 2

**Summary:**

This paper presents LeGS, a reinforcement learning (RL) framework for adaptive density control in 3D Gaussian Splatting (3DGS). Unlike existing heuristic strategies, a policy network determines operations for Gaussian primitive densification based on sensitivity-driven rewards. Although network-based densification introduces additional computational overhead, the authors mitigate this with an efficient closed-form solution that reduces complexity. Experiments on novel view synthesis datasets demonstrate consistent improvements in both reconstruction quality and efficiency.

**Compliance With Llm Reviewing Policy:**

Affirmed.

**Final Justification:**

Accept 5


The authors have resolved all my concerns. Therefore, I will maintain my positive score.

**Key Questions For Authors:**

- Since LeGS is implemented on top of FastGS, how would the method perform if applied to vanilla 3DGS instead? Could you clarify whether the reported gains primarily stem from the proposed learnable density control rather than FastGS framework?

- How does LeGS scale to very large or dynamic scenes? It would be helpful to validate this method on large-scale scenes, such as H3DGS dataset [1] and Zip-NeRF dataset [2], or dynamic scenes, such as iPhone dataset [3]. This does not necessarily require experimental validation, but conceptual or analytical evidence would strengthen the claim.

---
[1] Kerbl et al., A hierarchical 3d gaussian representation for real-time rendering of very large datasets, ACM ToG 2024
[2] Barron et al., Zip-nerf: Anti-aliased grid-based neural radiance fields, ICCV 2023
[3] Gao et al., Monocular Dynamic View Synthesis: A Reality Check, NeurIPS 2022

**Limitations:**

- The main limitation is that the RL-based density control primarily reformulates existing heuristics and depends on established RL pipelines, making the contribution appear incremental.
- Experimental validation of efficiency and scalability remains limited compared to quality metrics.

**Strengths And Weaknesses:**

## Strengths
- The paper clearly identifies the limitations of heuristic density control and introduces a RL–based paradigm.
- The sensitivity-based reward is well designed, efficiently computed, and enables high-quality reconstruction.
- Experimental results show strong improvements across diverse datasets, with ablation studies confirming robustness to varying Gaussian counts.

---

## Weaknesses

- Although the RL-based density control is novel, the main contribution lies in reformulating heuristics into RL, which may be incremental. Especially, the reward design is largely adapted from existing sensitivity analysis methods.
- Training stability and scalability to large-scale scenes (e.g., city-scale scenes) or dynamic scenes are not sufficiently explored.

---

> ### Author Rebuttal · Authors · 2026-03-31
>
> **[Q1] Performance of LeGS on vanilla 3DGS:**
>
> We sincerely thank the reviewer for the valuable comments. LeGS, when built on vanilla 3DGS, consistently improves all metrics across datasets by margins comparable to those reported in the paper (e.g., +0.53 dB PSNR on Tanks & Temples), while using significantly fewer Gaussians. These results demonstrate the genuine effectiveness of our method.
>
> | Dataset | Method | SSIM↑ | PSNR↑ | LPIPS↓ | #G↓ |
> | --- | --- | --- | --- | --- | --- |
> | Mip-NeRF 360 | 3DGS | 0.815 | 27.59 | 0.215 | 2.74 |
> |  | LeGS | **0.828** | **27.94** | **0.201** | **2.06** |
> | Tanks & Temples | 3DGS | 0.854 | 23.75 | 0.169 | 1.68 |
> |  | LeGS | **0.862** | **24.28** | **0.164** | **1.02** |
> | Deep Blending | 3DGS | 0.907 | 29.80 | 0.238 | 2.48 |
> |  | LeGS | **0.909** | **29.92** | **0.230** | **1.13** |
>
> **[Q2] Scalability to large-scale and dynamic scenes:**
>
> Thanks for insightful suggestion. In large-scale scene reconstruction, existing methods based on heuristic density control requires to partition scenes into hierarchical levels and optimize Level-of-Detail (LOD) structures independently. For instance, Octree-GS use different hyperparameters per level to control densification. By contrast, our LeGS can incorporate viewing distance and each Gaussian’s spatial region as inputs to the policy network, enabling joint optimization across hierarchy levels. We cannot perform experiments on it due to the limited time and will include them in the final paper.
>
> Nevertheless, in response to another reviewer, we conduct experiments to evaluate the zero-shot generalization capability of our LeGS.  Specifically, we conducted experiments on indoor scenes. Specifically, we trained the RL network of LeGS on Counter, Kitchen, and Bonsai scenes in the Mip-NeRF 360 dataset and directly applied it to Room in a zero-shot manner. The results below indicates the strong zero-shot generalization capability of LeGS. We surmise that a single policy network could also be used to optimize different hierarchical levels within the same scene, reducing manual tuning and yielding more coherent reconstructions in large-scale scenes.
>
> | Method | **SSIM↑** | **PSNR↑** | **LPIPS↓** |
> | --- | --- | --- | --- |
> | FastGS | 0.923 | 31.96 | 0.212 |
> | LeGS (zero-shot) | 0.927 | 32.14 | 0.199 |
>
> In dynamic scene reconstruction, 4DGS extends 3DGS by training a Gaussian Deformation Field Network to model temporal deformations, enabling dynamic scene representation. It retains the same Gaussian density control mechanism as 3DGS, allowing LeGS to be directly integrated into 4DGS for improved performance.
>
> Due to time constraints, we directly adapt LeGS to 4DGS for the Broom and Banana scenes in HyperNeRF and report PSNR. Nevertheless, since the density control process is learnable, temporal information can be naturally incorporated as additional input to the policy network, enabling action decisions at each time step and adaptive regulation of Gaussian density over time. This suggests a promising direction for future work.
>
> |  | Broom | Banana |
> | --- | --- | --- |
> | 4DGS | 22.0 | 28.0 |
> | 4DGS + LeGS | 22.4 | 28.2 |
>
> Although LeGS builds on existing components such as sensitivity analysis and PPO, its learnable density control for 3DGS is both practical and effective, providing a strong foundation for future extensions to large-scale and dynamic scene reconstruction.

---

> > ### Author Rebuttal · Reviewer_GTjf · 2026-04-04
> >
> > The authors have resolved all my concerns. Therefore, I will maintain my positive score.

---

### Official Review · Reviewer_yTYA · 2026-03-13

**Soundness:** 3
**Presentation:** 2
**Significance:** 3
**Originality:** 3
**Overall Recommendation:** 4
**Confidence:** 5

**Summary:**

In this paper, the author proposed a learned density control for 3D Gaussian Splatting. The designed framework applies reinforcement learning to optimize a policy network that generates operations to control the growth and elimination of Gaussians. A sensitivity metric is used as the reward for optimizing the policy network, while the author further derived a closed-form solution to accelerate the calculation of this reward metric. Experiment results on standard benchmarks in NeRF/3DGS research validate the effectiveness of the method.

**Compliance With Llm Reviewing Policy:**

Affirmed.

**Final Justification:**

Initially I'm learning to a weak reject, but after reading the author's response and comments from other reviewers, my major concerns are solved and i would like to increase my rating to weak accept.

**Key Questions For Authors:**

Please see the questions listed in the weaknesses section.

**Limitations:**

No. The paper includes only a very brief impact statement and does not meaningfully discuss methodological limitations.

**Strengths And Weaknesses:**

Strengths:
1. The idea of this work is well-motivated. This paper aims to address an important question, the density control of Gaussian reconstruction. Using RL to generate output actions of split, clone and prune is a reasonable and original direction for this problem.
2. The sensitivity-based reward is technically meaningful, and the reformulation of the sensitivity score into an efficient closed-form computation makes the overall learning framework much more practical.
3. The author validates this method on standard benchmarks of this field and shows that the proposed method improves the performance quantitatively across multiple datasets and scenes.
4. The writing is generally structured and understandable.

Weaknesses:
1. Some implementation details should be specified more clearly, such as the exact state inputs to the policy, the frequency of reward computation, the action execution details, and the full PPO/training hyperparameters. The paper gives the high-level pipeline, but reproducibility would be improved with a more complete description.
2. Although the paper proposes a learned method for density control on top of previous heuristic work, no theoretical guarantee or deeper analysis is provided regarding the convergence, stability, or optimality of the proposed RL-based control policy. Given that RL is a central part of the method, a stronger analysis would improve the paper’s quality and soundness.
3. I failed to find a report on the memory usage of the method in comparison with other baselines.

The presentation quality of this paper can still be improved by fixing a number of grammar and typo issues:
1. line 018 “recent works have investigates” -> should be “recent works have investigated.”
2. line 149 “rely on heuristic rules with extensive hyperparameter” -> should be “with extensive hyperparameters.”
3. line 110 “3DGS synthetic image via alpha-blending.” -> “3DGS synthesizes images via alpha-blending.”
4. “Follow 3DGS ..., we conduct experiments ...” -> should be “Following 3DGS ..., we conduct experiments ...” Also “as the efficiency metrics” should be singular: “as the efficiency metric.”
5. “despite employing higher number of Gaussians” -> should be “despite employing a higher number of Gaussians.”

Generally, the paper is readable, but it does need an throughout polish.


Overall, I find the paper promising and technically interesting. The motivation is clear, the method is reasonably novel, and the experiments support the usefulness of the approach. However, the paper would be stronger with cleaner presentation, more complete implementation details, and a more thorough analysis of the RL formulation. I encourage the authors to address the above issues to better meet the standard of an ICML paper.

---

> ### Author Rebuttal · Authors · 2026-03-31
>
> We sincerely thank the reviewer for pointing out the writing issues. We will correct all grammatical errors and typos and provide more detailed descriptions in the final paper. The code will also be fully open-sourced to ensure reproducibility.
>
> **[Q1] Clarification of implementation details for reproducibility:**
>
> Figure 2 shows the policy’s state inputs, including the gradients of each Gaussian’s position, opacity, scale, and principal spherical harmonics coefficients, along with the sensitivity score. The reward is computed at the same frequency as densification, i.e., every 100 iterations. PPO hyperparameters: $\gamma$=0.99, $\lambda$=0.95, clip ratio=0.2, epochs=2, n-step TD=2, LR 1e−3 $\to$ 1e−5. More details will be provided in the final paper.
>
> **[Q2] More Analysis of LeGS w.r.t. convergence, stability, and optimality of the RL-based density control policy:**
>
> Thank you for the insightful comment. Conducting a rigorous theoretical guarantee for the  convergence and global optimality of the proposed RL density control policy is quite challenging due to the non-convex property of 3DGS optimization and the complicated optimization of  PPO training.  In principle, heuristic density control relies on pre-defined thresholds, which result in uniform, low-dimensional decision boundaries. By contrast, our learnable RL paradigm enables fine-grained, state-dependent decisions, yielding adaptive boundaries across diverse scenes and resulting in more accurate and robust density control.
>
> We provide more empirical insights w.r.t. the convergence, stability, and optimality by conducting additional experiments. During training, at every 2,000 iterations from 3,000 to 15,000, we measure the proportion of RL-predicted actions that match the “locally optimal action” defined below and compare this with the FastGS heuristic. Herein, we define “locally optimal action” as follows: we sample 5% of the Gaussians and evaluate all four actions (maintain, clone, split, prune). For each action, we compute the PSNR improvement over the subsequent 50 iterations to identify the “locally optimal action”, and then measure how often the RL policy and FastGS match this choice.
>
> |  | 3000 | 5000 | 7000 | 9000 | 11000 | 13000 | 15000 |
> | --- | --- | --- | --- | --- | --- | --- | --- |
> | FastGS | 26.39% | 29.17% | 33.77% | 33.95% | 35.25% | 36.22% | 35.60% |
> | LeGS | 49.34% | 53.43% | 59.01% | 62.36% | 63.62% | 64.09% | 64.85% |
>
> The results show that LeGS consistently selects significantly more accurate  “locally optimal actions” than FastGS, which empirically demonstrates the superiority of the learnable density control via RL over the heuristic paradigm in terms of convergence, stability, and optimality.  We hope the proposed learnable density control paradigm in this work will inspire more promising approaches in this line of research.
>
> **[Q3] GPU Memory usage comparison with baseline methods:**
>
> Peak GPU memory usage (GB) is summarized as follows. Although the RL procedure introduces additional intermediate variables and rollout data, the policy network is lightweight, and such data are loaded onto the GPU only when needed. As a result, GPU memory usage remains comparable to that of baseline methods, with no significant overhead observed in practice.
>
> | Method | MipNerf360 | Tanks & Temples | Deep Blending |
> | --- | --- | --- | --- |
> | 3DGS-MCMC | 11.94 | 6.73 | 9.47 |
> | FastGS* | 8.99 | 7.80 | 6.01 |
> | LeGS | 12.57 | 9.6 | 8.12 |

---

> > ### Author Rebuttal · Reviewer_yTYA · 2026-04-03
> >
> > Some of my concerns are resolved in the rebuttal. However, I'm leaning towards keeping my current rating because I'm expecting theoretical analysis beyond empirical analysis on the superiority of the RL strategy in terms of convergence, stability, and optimality. Purely demonstrating with experiment results is insufficient for this work from my end.

---

> > > ### Author Response · Authors · 2026-04-04
> > >
> > > We thank the reviewer for the continued feedback.
> > >
> > > Prior RL theory usually guarantees **monotonic improvement across policy updates**, rather than global optimality [1]. For instance, TRPO provides a monotonic improvement lower bound: $\eta(\tilde\pi)\ge L_{\pi}(\tilde\pi)-\frac{4\epsilon\gamma}{(1-\gamma)^2}\alpha^2$, implying that **one-step improvement** can be guaranteed via a surrogate lower bound [1].
> > >
> > > Consistent with this perspective, below we provide an approximate bound on the deviation of our one-step decision from the global (one-step) optimum.
> > >
> > > Recall the Sensitivity Score in our paper, which measures each Gaussian’s contribution at time step $t$:
> > > $\mathrm{Sen}^t_{i}=\sum\_{p\in pix(\mathcal{G}^t_i)}\Big(|C^t_{-i}(p)-C\_{gt}(p)|-|C^t(p)-C_{gt}(p)|\Big)$,
> > > where $C^t(p)$ and $C_{-i}^t(p)$ are the rendered colors using all Gaussians and excluding $\mathcal{G}_i$, $C\_{gt}(p)$ is the ground-truth color, and $pix(\mathcal{G}^t_i)$ denotes pixels covered by $\mathcal{G}^t_i$ ’s 2D footprint.
> > >
> > > We also define the Sensitivity Reward, measuring the sensitivity gain from a parent Gaussian to its offspring after action $a$: $R^t_i=\Big(\sum_{j\in child(\mathcal{G}^t_i)}\mathrm{Sen}_j^{t+1}\Big)-\mathrm{Sen}^t_i$.
> > >
> > > Based on these definitions, we define the reconstruction loss for pixel $p$ as an optimality measure:
> > > $\ell_p(C)=|C-C\_{gt}(p)|$.
> > > Maximizing the Sensitivity Reward is approximately equivalent to maximizing one-step optimality as follows.
> > >
> > > For action $a$, let the rendered color be $C_a^{t+1}(p)$ after applying $a$ to $\mathcal{G}^t_i$, and let the rendering after removing the offspring group be $B(p)$*.* Since $a$ only replaces the parent with its offspring, we have $B(p)=C_{-i}^t(p)$.
> > >
> > > Define the offspring-group sensitivity similarly to
> > > $\mathrm{Sen}^t_i$:
> > >
> > > $\mathrm{Sen}^{t+1}_{\mathrm{child},p}(a)=\ell_p(B(p))-\ell_p(C_a^{t+1}(p))$
> > >
> > > Then the exact sensitivity gain from the parent Gaussian to its offspring group can be computed as:
> > >
> > > $R^{\star}_{i,p}(a)=\mathrm{Sen}^{t+1}\_{\mathrm{child},p}(a) - \mathrm{Sen}^t\_{i,p} = \ell_p(C^t(p)) -\ell_p(C_a^{t+1}(p))$
> > >
> > > Since $R^\star_{i,p}(a)$ is determined solely by $\ell_p(C_a^{t+1}(p))$, maximizing $R^\star_{i,p}(a)$ is exactly equivalent to minimizing $\ell_p(C_a^{t+1}(p))$. **Hence, maximizing the exact sensitivity gain directly yields a one-step optimum.**
> > >
> > > Accordingly, our Sensitivity Reward $R^t_i$ serves as a tractable surrogate for the exact gain
> > > $R^\star_{i,p}(a)$:
> > > $R_{i,p}(a)=\sum_{j\in child(\mathcal{G}i^t)}\mathrm{Sen}^{t+1}\_{j,p}-\mathrm{Sen}^t\_{i,p}=R^\star_{i,p}(a)+\varepsilon_{i,p}(a)$,
> > >
> > > where $\varepsilon_{i,p}(a)$ is the interaction error induced by $\alpha$-blending, **which is zero when offspring contributions are pixel-wise non-interacting and small when interactions are weak.**
> > >
> > > **Prove one-step local approximate optimality:**
> > > Assume $|\varepsilon_{i,p}(a)|\le\delta$ for all $a$, and let $a^\star=\arg\max_a R^\star_{i,p}(a)$ and  $\hat a=\arg\max_a R_{i,p}(a)$. Then:
> > >
> > > 1. **Exact recovery under a margin condition**: if $R^\star_{i,p}(a^\star)-\max_{a\neq a^\star}R^\star_{i,p}(a) > 2\delta$, then  $\hat a=a^\star$.
> > > 2. **Approximate optimality without a margin condition**: $R^\star_{i,p}(a^\star)-R^\star_{i,p}(\hat a)\le 2\delta$, equivalently $\ell_p(C_{\hat a}^{t+1}(p))\le \ell_p(C_{a^\star}^{t+1}(p))+2\delta$.
> > >
> > > Thus, our chosen action is at most $2\delta$-suboptimal for the **one-step local** decision at the pixel level.
> > >
> > > **Prove Convergence (image-level):**
> > > Let the reconstruction loss for a whole image be:
> > > $L_t=\sum_{p\in pix}\ell_p(C^t(p))$.
> > > The exact global one-step sensitivity gain for an image is:
> > > $R^\star_{i,t}(a)=L_t-L_{t+1}$.
> > > Assume the implemented reward satisfies $R^t_i(a)-R_i^{\star,t}(a)|\le\delta_g$. If we accept an edit only when $R^t_i(\hat a_t)\ge \tau$, then $R_i^{\star,t}(\hat a_t)\ge \tau-\delta_g$. Therefore, if $\tau>\delta_g$, each accepted edit strictly decreases the global loss:
> > > $L_{t+1}\le L_t-(\tau-\delta_g)$.
> > > Since $L_t\ge 0$, the loss over the accepted-edit sequence is monotonically decreasing and bounded below, hence convergent, and the number of accepted edits is finite: $T\le \frac{L_0}{\tau-\delta_g}$. For $\tau=0$, we only obtain $L_{t+1}\le L_t+\delta_g$, which gives a local tolerance guarantee but not monotone convergence by itself.
> > >
> > > In summary, we do not claim global optimality. Instead, our reward gives a rigorous **one-step local approximate-optimality** view. With bounded reward error and a positive acceptance margin, the editing process yields **monotone loss decrease** and **finite-step termination**.
> > >
> > > Finally, we do not aim to find a globally optimal density-control solution. Instead, we take **a first step toward reframing it as a learnable problem** and **validating its experimental feasibility**. We hope this inspires future work on learnable density control.
> > >
> > > [1] Schulman, John, et al. "Trust region policy optimization." International conference on machine learning. PMLR, 2015.

---

### Official Review · Reviewer_gKyZ · 2026-03-16

**Soundness:** 3
**Presentation:** 3
**Significance:** 3
**Originality:** 3
**Overall Recommendation:** 4
**Confidence:** 3

**Summary:**

The paper proposes LeGS, a learnable alternative to heuristic density control in 3D Gaussian Splatting. It formulates densification and pruning as a discrete action policy (maintain, clone, split, prune) trained with reinforcement learning, using a sensitivity-based reward that quantifies each Gaussian’s marginal contribution to reconstruction quality via an efficient closed-form computation that avoids re-rendering. Experiments on Mip-NeRF360, Tanks & Temples, and Deep Blending show consistent quality improvements with competitive Gaussian counts and minimal runtime overhead relative to strong baselines.

**Compliance With Llm Reviewing Policy:**

Affirmed.

**Key Questions For Authors:**

[Q1]
Eq. 8, C_{-i}(p) has (1-alpha_i) in denom, so when opacity approaches 1, this diverges, ?
Any discussion on clamping or masking, or how frequently does this occur during training?
Since new Gaussians are spawned and old ones pruned every  ~100 iterations, the number of active agents changes dynamically. But how does PPo handle this? Or even more important, does the KL constraint remain meaningful across varying policy support sets?
Some quantitative evidence is needed here, could be histogram of alpha_i values during training?

[Q2]
As I understand, LeGS has the same FastGS’ rendering backend.
Can authors ablate the LeGS policy on vanilla 3DGS, purely to isolate the gains from the RL policy and not being inherited rendering optimizations?
If the policy still outperforms vanilla 3DGS by the reported margin, then policy contribution from backend contribution can be isolated. Else, this contribution needs to be revised.

[Q3]
Can authors discuss and perhaps also cite (probably concurrent), RLGS (arXiv:2508.04078) and also Perceptual-GS (ICML’25), and especially the differentiation from their shared sensitivity-score intuition?
Another relevant paper is GDAGS (ICLR’26), which also addresses density control using a Gradient Coherence ratio. Can authors clarify if LeGS’ policy state vector includes any directional gradient features? To me current state features and the sensitivity score are all magnitudes or scalar

[Q4]
It would be only fair to compare against 3DGS-MCMC in Table 4, given that LeGS introduces a policy network (forward + backward through an MLP on potentially millions of Gaussians).
Authors should include a full training time profiling, especially since the RL reward is computed every 100 iterations.

[Q5]
Please include PPO hyperparameters, clip epsilon, GAE discount gamma and lamba.
Also, beyond the aggregated results, is it possible to include per-scene results, or at least find scenes where the heuristic baseline wins. I think this should support the strong scene-adaptive resource allocation claim.
More on the failure case, it is unclear what does the policy network do when trained on insufficient views or when the sensitivity reward cannot be relied upon, near alpha_i ~ 1?

**Limitations:**

Mostly included, but please see [Q2] above.

**Strengths And Weaknesses:**

**Strengths**
1. Alpha blending closed-form derivation is an algorithmic contribution and follows from the linearity of transmittance decomposition.

2. Sensitivity maps, PPO, MLP policy networks are all standard. To me this work’s novelty lies in recognizing that density control is a valid RL formulation (discrete action selection).

3. The paper is generally well-structured. The problem motivation is clear, and the contrast between the heuristic paradigm, Eq. 4 and the learned paradigm, Eq. 12-17 is clear.


**Weaknesses**
Please see the questions below.

---

> ### Author Rebuttal · Authors · 2026-03-31
>
> **[Q1] Numerical stability of** $C_{-i}(p)$ **and PPO behavior under dynamic Gaussian sets:**
>
> We thank the reviewer for the meticulous evaluation. Eq. (8) removes the contribution of Gaussian $\mathcal{G}_i$ from Eq. (7). When $\alpha_i = 1$, $(1 - \alpha_i)$ results in $T_k = 0$ for all $k > i$, so $\mathcal{G}_i$ becomes the last effective Gaussian. The second term in Eq. (8) thus vanishes, avoiding divergence. In practice, we cap the last effective Gaussian’s opacity at 0.99 and ignore subsequent ones for numerical stability.
>
> The dynamic number of Gaussians does not invalidate PPO. We use a shared per-Gaussian policy with a fixed action space, independent of the number of active Gaussians. PPO’s clipping/KL is applied per state, where the Gaussian count is fixed; it varies only across states (i.e., across timesteps), so the constraint holds.
>
> Due to the space limitation, we briefly report the $\alpha$ distribution in the bicycle scene after 7k and 30k iterations, which indicates that the disregarded Gaussians in [0.99, 1.00] account for only a negligible fraction.
>
> | Iteration | [0.00, 0.10] | [0.10, 0.50] | [0.50, 0.99] | [0.99, 1.00] |
> | --- | --- | --- | --- | --- |
> | 7000 | 22.82% | 68.30% | 8.84% | 0.02% |
> | 30000 | 14.48% | 50.91% | 32.06% | 1.34% |
>
> **[Q2] LeGS built on vanilla 3DGS:**
>
> Thanks a lot for the valuable comments. When built on vanilla 3DGS, LeGS consistently improves all metrics across datasets, with gains comparable to LeGS based on FastGS, demonstrating the effectiveness of our LeGS.
>
> | Dataset | Method | SSIM↑ | PSNR↑ | LPIPS↓ | #G↓ |
> | --- | --- | --- | --- | --- | --- |
> | Mip-NeRF 360 | 3DGS | 0.815 | 27.59 | 0.215 | 2.74 |
> |  | LeGS | **0.828** | **27.94** | **0.201** | **2.06** |
> | Tanks & Temples | 3DGS | 0.854 | 23.75 | 0.169 | 1.68 |
> |  | LeGS | **0.862** | **24.28** | **0.164** | **1.02** |
> | Deep Blending | 3DGS | 0.907 | 29.80 | 0.238 | 2.48 |
> |  | LeGS | **0.909** | **29.92** | **0.230** | **1.13** |
>
> **[Q3] Relation to concurrent works:**
>
> Thanks for pointing us to the concurrent works.
>
> While **RLGS** employs RL to learn global hyperparameters (e.g., learning rates), it still follows a heuristic density-control paradigm, in contrast to our LeGS which learns density control via RL.
>
> **Perceptual-GS** also relies on heuristic density control, defining image-structure cues (e.g., edges) as a sensitivity for an auxiliary rule. In contrast, our LeGS introduces a Sensitivity Score for fine-grained RL supervision based on occlusion analysis.
>
> Similarly, **GDAGS** introduces the directional consistency, constrained by gradient coherence, as a new heuristic rule.  In comparison, LeGS directly incorporates vector-valued Gaussian gradients (e.g., xyz, scaling, opacity, SH) into the RL states, offering more explicit directional signals.
>
> We have included the comparison with Perceptual-GS in Table 1 in the paper. Direct experimental comparisons with RLGS and GDAGS are not currently feasible because their code has not been released; we will add such comparisons once implementations become available.
>
> **[Q4] Full training-time comparison with 3DGS-MCMC:**
>
> We summarize the training-time comparison with 3DGS-MCMC as follows (D&P: densification and pruning; Reward: total reward computation; RL: reinforcement learning). 3DGS-MCMC incurs significant overhead due to per-iteration covariance computation across all Gaussians, leading to longer training times.
>
> | Dataset | Method | Total | Render | D&P | Reward | RL |
> | --- | --- | --- | --- | --- | --- | --- |
> | MipNerf360 | 3DGS-MCMC | 2639 | 383 | 6 | N/A | N/A |
> |  | LeGS | 745 | 128 | 58 | 46 | 63 |
> | Tanks&Temples | 3DGS-MCMC | 1385 | 231 | 4 | N/A | N/A |
> |  | LeGS | 569 | 103 | 43 | 36 | 52 |
> | DeepBlending | 3DGS-MCMC | 2244 | 276 | 5 | N/A | N/A |
> |  | LeGS | 483 | 79 | 38 | 28 | 47 |
>
> **[Q5] PPO hyperparameters, per-scene performance, and failure cases:**
>
> PPO hyperparameters: $\gamma=0.99$, $\lambda=0.95$, clip ratio=0.2, epochs=2, n-step TD=2, LR 1e−3 $\to$ 1e−5. More details will be provided in the camera-ready version.
>
> Due to the space limit, we report most per-scene results on Mip-NeRF 360 (full version will be included in the final paper). In treehill (insufficient views), LeGS slightly reduces PSNR but improves SSIM/LPIPS, indicating better structure and texture preservation. For the sensitivity reward, no division-by-zero occurs in practice even when $\alpha$ is close to 1, as discussed in [Q1].
>
> | Scene | PSNR ↑ | SSIM ↑ | LPIPS ↓ |
> | --- | --- | --- | --- |
> | bicycle | 25.740 (+0.503) | 0.800 (+0.039) | 0.168 (-0.064) |
> | garden | 28.066 (+0.369) | 0.881 (+0.009) | 0.090 (-0.010) |
> | stump | 27.376 (+0.408) | 0.811 (+0.028) | 0.176 (-0.051) |
> | treehill | **22.797 (-0.049)** | **0.658 (+0.029)** | **0.272 (-0.093)** |
> | room | 32.482 (+0.519) | 0.931 (+0.008) | 0.193 (-0.019) |
> | kitchen | 32.673 (+0.705) | 0.936 (+0.002) | 0.114 (-0.000) |
> | bonsai | 33.532 (+0.788) | 0.953 (+0.006) | 0.177 (-0.008) |

---

> > ### Author Rebuttal · Reviewer_gKyZ · 2026-04-03
> >
> > Thank the authors for their detailed rebuttal and the additional experiments provided.

---

### Decision · Program_Chairs · 2026-04-30

**Decision:**

Accept (regular)

**Comment:**

All reviewers are positive about this submission and agree on acceptance. The paper introduces a reinforcement learning formulation for density control in 3D Gaussian Splatting, which is recognized as a meaningful and technically sound contribution. The method demonstrates promising improvements over strong baselines such as FastGS, while maintaining reasonable training overhead and memory usage.

I do not observe any critical issues, and the work provides valuable insights that are likely to benefit the community.